# Explosive mutation accumulation triggered by heterozygous human Pol ε proofreading-deficiency is driven by suppression of mismatch repair

Karl P Hodel[1†], Richard de Borja[2†], Erin E Henninger[1†‡], Brittany B Campbell[3,4], Nathan Ungerleider[5], Nicholas Light[2], Tong Wu[5], Kimberly G LeCompte[1§], A Yasemin Goksenin[1#], Bruce A Bunnell[6,7], Uri Tabori[2,3,8], Adam Shlien[2,9,10], Zachary F Pursell[1,11]*

[1]Department of Biochemistry and Molecular Biology, Tulane University School of Medicine, New Orleans, United States; [2]Program in Genetics and Genome Biology, The Hospital for Sick Children, Toronto, Canada; [3]The Arthur and Sonia Labatt Brain Tumour Research Centre, The Hospital for Sick Children, Toronto, Canada; [4]Institute of Medical Science, Faculty of Medicine, University of Toronto, Toronto, Canada; [5]Department of Pathology, Tulane University School of Medicine, New Orleans, United States; [6]Department of Pharmacology, Tulane University School of Medicine, New Orleans, United States; [7]Tulane Center for Stem Cell Research and Regenerative Medicine, Tulane University School of Medicine, New Orleans, United States; [8]Division of Hematology/Oncology, The Hospital for Sick Children, Toronto, Canada; [9]Department of Paediatric Laboratory Medicine, The Hospital for Sick Children, Toronto, Canada; [10]Department of Laboratory Medicine and Pathobiology, University of Toronto, Toronto, Canada; [11]Tulane Cancer Center, Tulane University School of Medicine, New Orleans, United States

*For correspondence:
zpursell@tulane.edu

†These authors contributed equally to this work

Present address: ‡Sorbonne Universités, UPMC Univ Paris 06, CNRS, UMR8226, Laboratoire de Biologie Moléculaire et Cellulaire des Eucaryotes, Institut de Biologie Physico-Chimique, Paris, France; §Louisiana State University, Baton Rouge, United States; #Department of Pediatrics, University of California, San Francisco, United States

Competing interests: The authors declare that no competing interests exist.

**Abstract** Tumors defective for DNA polymerase (Pol) ε proofreading have the highest tumor mutation burden identified. A major unanswered question is whether loss of Pol ε proofreading by itself is sufficient to drive this mutagenesis, or whether additional factors are necessary. To address this, we used a combination of next generation sequencing and in vitro biochemistry on human cell lines engineered to have defects in Pol ε proofreading and mismatch repair. Absent mismatch repair, monoallelic Pol ε proofreading deficiency caused a rapid increase in a unique mutation signature, similar to that observed in tumors from patients with biallelic mismatch repair deficiency and heterozygous Pol ε mutations. Restoring mismatch repair was sufficient to suppress the explosive mutation accumulation. These results strongly suggest that concomitant suppression of mismatch repair, a hallmark of colorectal and other aggressive cancers, is a critical force for driving the explosive mutagenesis seen in tumors expressing exonuclease-deficient Pol ε.
DOI: https://doi.org/10.7554/eLife.32692.001

## Introduction

Human cancers share common features of genome instability and mutagenesis (*Hanahan and Weinberg, 2011*) that are the sources of the $10^3$ to $10^6$ somatic mutations observed in the genomes of most types of adult tumors (*Stratton, 2011*; *Wheeler and Wang, 2013*). The total mutation burden in a tumor is the result of multiple mutational pathways operating within the cells at varying rates

**eLife digest** New cells are made when an existing cell divides in two. Each time a cell divides, it duplicates its DNA so that each new cell inherits a complete copy. Molecular machines called DNA polymerases make these DNA copies. The main DNA polymerases, known as delta and epsilon, can "proofread" the new DNA, which ensures that the genetic information stored in the DNA is correctly copied. Cells also use another system, called mismatch repair, to catch any errors that get missed by the polymerases.

Cancer cells contain many mutations in genes that regulate the growth and production of new cells, which is why cancers grow out of control and produce tumors. Research shows that many cancer cells with high numbers of mutations have lost their proofreading ability. Yet it is not clear if the loss of proofreading is enough to cause cancers, or if other systems, such as mismatch repair, must also be defective.

Hodel, de Borja, Henninger et al. examined human cells grown in the laboratory to understand the importance of proofreading in cancer. It turns out that even the partial loss of polymerase epsilon proofreading could lead to distinctive mutations. Yet, these mutations were repaired by mismatch repair, so they actually are only found in cells when mismatch repair is also defective. This result demonstrates that the lack of proofreading is not enough to cause a large number of mutations. These cancers only happen when other systems are damaged too.

These new findings add to the current understanding of the origins of mutations in cancers and how mutations accumulate over time. It should lead scientists to further investigate the patterns of mutations that happen in the absence of proofreading. It may also enhance our knowledge of proofreading-deficient cancers.

DOI: https://doi.org/10.7554/eLife.32692.002

over time. This can complicate attempts to assign the relative contributions of each pathway to the mutation spectrum of a tumor. One essential tool to our understanding of how mutations accumulate and influence tumor progression is using computational means to extract multiple individual signatures from many tumor genomes (*Alexandrov et al., 2013a*; *Alexandrov and Stratton, 2014*; *Haradhvala et al., 2016*). This is proving to be instrumental in resolving the relative extents to which pathways contribute to the ultimate mutation spectrum in a tumor (*Nik-Zainal et al., 2016*; *Roberts et al., 2013*). Comparing these tumor mutation signatures to those generated in experimental cell lines is another critical tool to understanding the relative rates and causality of mutation acquisition (*Fox et al., 2016*; *Helleday et al., 2014*). Traditionally, these measurements have relied on assays using reporter genes, which necessarily look at a tiny fraction of the genome and may miss global contributions to genome instability. Advances in next generation sequencing now allow for detailed genome-wide analyses of mutation accumulation over defined periods of cellular growth. Since each nucleotide in the genome is subject to the three major determinants of replication fidelity - nucleotide selection, proofreading and mismatch repair (MMR) - during every round of replication, tumors and cells with defects in replication fidelity are uniquely poised to address these issues.

Proofreading defects are now known to occur in a wide variety of tumors, with significant enrichment in colorectal and endometrial tumors (*Cancer Genome Atlas Network, 2012*; *Kandoth et al., 2013*; *Heitzer and Tomlinson, 2014*; *Rayner et al., 2016*). Mutations in DNA polymerase (Pol) ε cluster in the exonuclease proofreading domain and the tumors are clinically characterized by several criteria, including being ultrahypermutated, having a unique mutation spectrum, containing a heterozygous Pol ε mutation with no evidence of loss of heterozygosity (LOH) and being microsatellite stable (MSS) (*Briggs and Tomlinson, 2013*; *Church et al., 2013*; *Palles et al., 2013*; *Zhao et al., 2013*; *Henninger and Pursell, 2014*; *Shinbrot et al., 2014*; *Shlien et al., 2015*; *Barbari and Shcherbakova, 2017*). Whole genome and whole exome analyses of tumors have been the primary means to establish the ultrahypermutated (>100 Mutations per megabase) unique mutational signature that distinguish Pol ε tumors from other cancers (*Alexandrov et al., 2013a*; *Alexandrov and Stratton, 2014*; *Shinbrot et al., 2014*; *Shlien et al., 2015*; *Alexandrov et al., 2013b*; *Campbell et al., 2017*). While there is a rich history of studies on the effects of exonuclease defects on mutagenesis in model

organisms, the extent to which Pol ε proofreading-deficiency by itself drives each of these criteria remains poorly understood.

It is clear from studies in model organisms that complete, biallelic inactivation of Pol ε proofreading activity causes mutagenesis and carcinogenesis in model organisms, where mutation rates have been precisely measured using reporter genes. For example, mutation rates are increased in haploid or diploid yeast strains expressing only proofreading-deficient alleles of Pols ε (*Morrison et al., 1991*; *Morrison and Sugino, 1994*; *Shcherbakova et al., 2003*) or δ (*Morrison et al., 1993*; *Simon et al., 1991*; *Herr et al., 2011a*). These rates are further elevated when combined with defects in mismatch repair, indicating that these errors are made during replication (*Morrison and Sugino, 1994*; *Tran et al., 1999*; *Tran et al., 1997*; *Kennedy et al., 2015*). In mouse models, homozygous inactivation of both copies of either Pol ε or δ exonuclease activity (Pol ε$^{exo-/exo-}$ or Pol δ$^{exo-/exo-}$) causes increased mutation rates and cancer (*Albertson et al., 2009*; *Goldsby et al., 2002*; *Goldsby et al., 2001*). Interestingly, their tumor spectra are different, with gastrointestinal tumors predominant in Pol ε$^{exo-/exo-}$ mice while thymic lymphomas are the major tumor in Pol δ$^{exo-/exo-}$ mice.

However, mice with a heterozygous inactivation of a single Pol ε proofreading allele (the monoallelic Pol ε$^{wt/exo-}$ genotype) fail to develop tumors when mismatch repair is functional (*Albertson et al., 2009*). The equivalent diploid heterozygous Pol ε exonuclease mutant in yeast is also a mutator, but the effect is modest and partially dominant to the wild type allele and lacks the unique mutation spectrum seen in human tumors (*Morrison and Sugino, 1994*; *Shcherbakova et al., 2003*; *Morrison et al., 1993*; *Kane and Shcherbakova, 2014*). These results raise critical questions as to the source of the unique, ultrahypermutant phenotype in human tumors with heterozygous Pol ε exonuclease-deficiency.

Mismatch repair is responsible for the recognition and removal of replication errors and deficiencies in this activity cause genome instabilities that can lead to cancer (*Kunkel and Erie, 2005*; *Li, 2008*; *Jiricny, 2013*; *Modrich, 2006*). Mismatch repair is normally an extremely efficient process, correcting more than 99% of replication errors. However, genome-wide studies have recently shown that MMR efficiencies can vary by over two orders of magnitude and are influenced by a number of factors, including the strand on which the mismatch occurs, the polymerase that made the error, the nature of the mismatch, local sequence context, distance from the origin and replication timing (*Hawk et al., 2005*; *Hombauer et al., 2011*; *Lujan et al., 2014*; *Lujan et al., 2012*; *Supek and Lehner, 2015*). Patients with biallelic mismatch repair disorder (bMMRD) have biallelic germline inactivating mutations in a mismatch repair gene and are completely lacking mismatch repair and develop a number of early-onset tumors in which microsatellite instability (MSI) is readily detectable (*Durno et al., 2017*; *Wimmer et al., 2014*). A subset of these patients acquires a later somatic mutation in a single allele of Pol ε, leading to very aggressive tumor development. Mutation rates from these Pol ε$^{wt/exo-}$ MMR$^{-/-}$ tumors have been estimated on the order of several hundred per genome duplication (*Shlien et al., 2015*). This is consistent with results from model systems as mice with the equivalent genotype (heterozygous Pol ε$^{wt/exo-}$ combined with homozygous MMR$^{-/-}$) develop tumors within 1–2 months (*Treuting et al., 2010*). The equivalent yeast strains are strong mutators as well (*Shcherbakova et al., 2003*; *Morrison et al., 1993*; *Kennedy et al., 2015*).

However, since sporadic POLE tumors are generally microsatellite stable, the role of MMR in Pol ε proofreading-deficiency in the development of these MSS tumors remains a critical unanswered question. Whether MMR and POLE defects together are required for ultramutation, elevated mutation rates or for establishing the unique mutation signature is unknown. Understanding how MMR function or dysfunction affects proofreading-dependent mutagenesis is essential to understanding the mechanisms of mutagenesis during cancer development.

In the current study, we constructed a human cell line model system to address the roles of Pol ε proofreading in driving the clinical characteristics that define Pol ε tumors. Critically, we used a targeted knock-in approach to inactivate one copy of Pol ε 3'−5' exonuclease activity, since human tumors contain heterozygous, monoallelic Pol ε mutations. Using mutation rates measured at a reporter gene in combination with whole-exome and whole-genome sequencing we found a rapid accumulation of large numbers of Pol ε-specific mutations in mismatch repair-deficient cells. This confirms results suggested by observations in Pol ε mutant bMMRD tumors. We further show that mismatch repair is able to suppress exonuclease-deficient Pol ε-induced mutation rates back to wild type levels using a combination of reporter gene and whole-exome sequencing (WES). These results support the idea that additional unique features beyond a single exonuclease active site inactivation

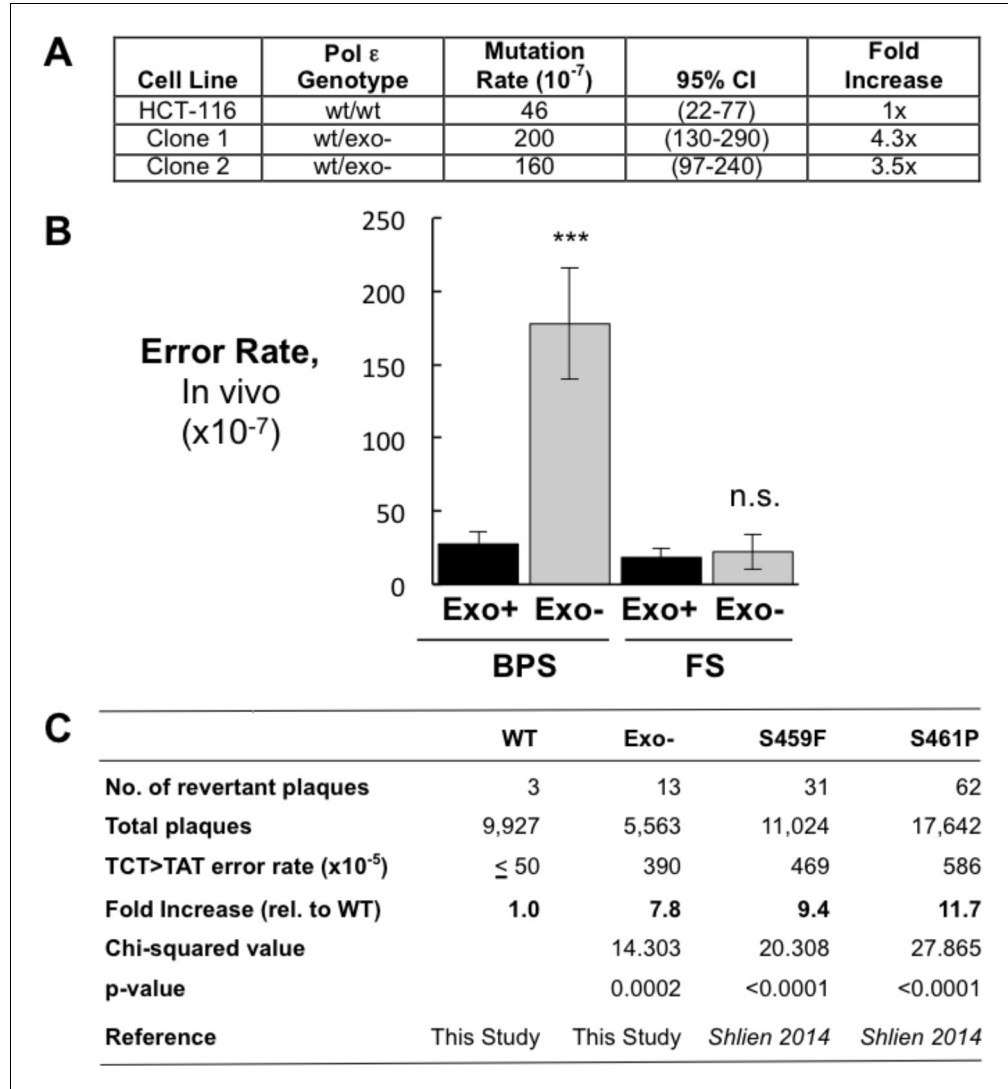

**Figure 1.** Heterozygous inactivation of Pol ε proofreading causes an increase in specific base pair substitutions. (A) Mutation rates were measured using the fluctuation assay at the HPRT1 locus by resistance to 6-thioguanine. Mutation rates and 95% confidence intervals were measured by fluctuation analysis as described in the Methods using the Ma-Sandri-Sarkar Maximum Likelihood Estimator. Twelve independent isolates of both the parental (wt/wt) cell line and two independently derived clones of the heterozygous cell lines (wt/exo-) were used. All cell lines were mismatch repair-deficient. P-values for Clones 1 and 2 (p=0.0017 and p=0.008, respectively) were calculated using an unpaired t-test relative to wt/wt. Mutation rates for Clone 1 and Clone 2 were not significantly different from one another (p=0.4727). (B) Error rates for base pair substitutions (BPS) and small insertion/deletion frameshift mutations (FS) were calculated using the mutation rate data from *Figure 1A*. Exo + BPS Error Rate = $27.6 \times 10^{-7}$, SEM = $8.48 \times 10^{-7}$, n = 12; Exo- BPS Error Rate = $178 \times 10^{-7}$, SEM = $37.8 \times 10^{-7}$, n = 8; p=0.0002. Exo + FS Error Rate = $18.4 \times 10^{-7}$, SEM = $5.73 \times 10^{-7}$, n = 8; Exo- FS Error Rate = $22.2 \times 10^{-7}$, SEM = $12.1 \times 10^{-7}$, n = 1; p=0.7759. Error rate data shown for Exo- is from Clone 1 (See *Figure 1A*). The HPRT1 ORF was sequenced from independently derived isolates of 6-TG resistant clones (these included 20 mismatch repair-deficient Pol ε[wt/wt] and 25 mismatch repair-deficient Pol ε[wt/exo-] clones; see Materials and methods). Sequence changes used to calculate error rates are in *Figure 1—source data 2*. \*\*\*p<0.001; n.s., p>0.05. (C) Errors rates were calculated using a lacZ reversion substrate that reverts via TCT→TAT transversion. P values were calculated using chi-square tests with Yates correction. Error rates are the averages of two experiments, each conducted with independent DNA and enzyme preparations for each construct tested. ≤indicates the value is a maximal estimate as it is identical to the assay background.

DOI: https://doi.org/10.7554/eLife.32692.003

The following source data and figure supplements are available for figure 1:

*Figure 1 continued on next page*

*Figure 1 continued*

**Source data 1.** Pol ε rAAV targeting efficiencies in human HCT-116 cells.
DOI: https://doi.org/10.7554/eLife.32692.006
**Source data 2.** HPRT1 mutations sequenced from 6-thioguanine resistant Pol ε wt/exo- and Pol ε wt/wt HCT116 cells.
DOI: https://doi.org/10.7554/eLife.32692.007
**Figure supplement 1.** Generation of exonuclease-deficient Pol ε human cell lines by gene targeting.
DOI: https://doi.org/10.7554/eLife.32692.004
**Figure supplement 2.** Southern blot of parental (HCT116) and knock-in clone (HCT116-Polε<sup>wt/exo-</sup>) after Cre-mediated excision.
DOI: https://doi.org/10.7554/eLife.32692.005

are helping facilitate the massive mutation acquisition seen in microsatellite stable tumors containing mutant Pol ε.

## Results

### Inactivation of Pol ε proofreading causes a mutator phenotype in human cells

Tumors with mutations in the exonuclease domain of POLE are generally microsatellite stable and show no or low loss of heterozygosity, suggesting that inactivation of exonuclease activity in one allele is sufficient to drive mutagenesis and tumor development, though this has not been directly tested previously. To test whether inactivation of a single allele of Pol ε proofreading was sufficient to cause a mutator phenotype in human cells, we used recombinant adenoassociated virus (rAAV)-mediated gene targeting to engineer a diploid human cell line to express one allele of Pol ε with the D275A/E277A double substitution (*Figure 1—figure supplements 1–2*; *Figure 1—source data 1*). We chose the D275A/E277A mutation because it inactivates exonuclease proofreading in vitro (*Shcherbakova et al., 2003*; *Korona et al., 2011*). The parental cell line, HCT-116, is constitutively mismatch repair-deficient due to an inactivating mutation in Mlh1, thus allowing us to first define the contributions of proofreading deficiency separately to mutagenesis. We then measured the mutation rate at the hypoxanthine-guanine phosphoribosyltransferase (HPRT1) locus using 6-thioguanine (6-TG) resistance and a fluctuation assay. The measurements were repeated in clones derived from independent exonuclease-deficient (exo-) allele integration events. A moderate mutator effect was seen in Pol ε<sup>wt/exo-</sup> heterozygotes (*Figure 1A*), indicating the exo- allele was partially dominant over the endogenous exo + allele, similar to what is seen in a mismatch repair-deficient diploid cell line heterozygous for a Pol ε proofreading mutation, *pol2-4/+pms1/pms1* (*Pavlov et al., 2004*). Mutation rates were not measured in cells from the comparable heterozygous Pol ε<sup>wt/exo-</sup> mice lacking mismatch repair (*Albertson et al., 2009*).

To begin measuring the effect of inactivating a single Pol ε exonuclease allele on mutation rates in cells, we sequenced the HPRT1 gene from twenty and twenty-five independently derived 6-TG<sup>R</sup> (and thus HPRT1 mutant) clones from mismatch repair-deficient Pol ε<sup>wt/wt</sup> and Pol ε<sup>wt/exo-</sup> cells, respectively (*Figure 1—source data 2*). This allowed comparison to previously measured mutation rates from different groups using the same parental cell line. Mutation rates from the Pol ε<sup>wt/wt</sup> cells were similar to the spontaneous mutation rates reported by three previous studies (*Bhattacharyya et al., 1995*; *Glaab and Tindall, 1997*; *Ohzeki et al., 1997*). These results suggest that the baseline rates of mutagenesis are an accurate measure of comparison for the Pol ε<sup>wt/exo-</sup> cell lines.

The increase in mutation rate seen in the Pol ε<sup>wt/exo-</sup> mismatch repair-deficient cells was primarily due to base pair substitutions (*Figure 1B*). Frameshift error rates did not change, in agreement with previous findings in vitro that Pol ε proofreading primarily strongly corrects base-base mispairs with little effect on frameshift fidelity (*Korona et al., 2011*). However, the number of mutational events scored by this method is insufficient to make statistical claims regarding individual mutations, reinforcing the need for genome sequencing to examine mutations in all possible sequence contexts.

Using an in vitro lacZ reversion substrate that specifically measures TCT→TAT transversions (*Shinbrot et al., 2014*; *Shlien et al., 2015*), the D275A/E277A mutant made these errors at a

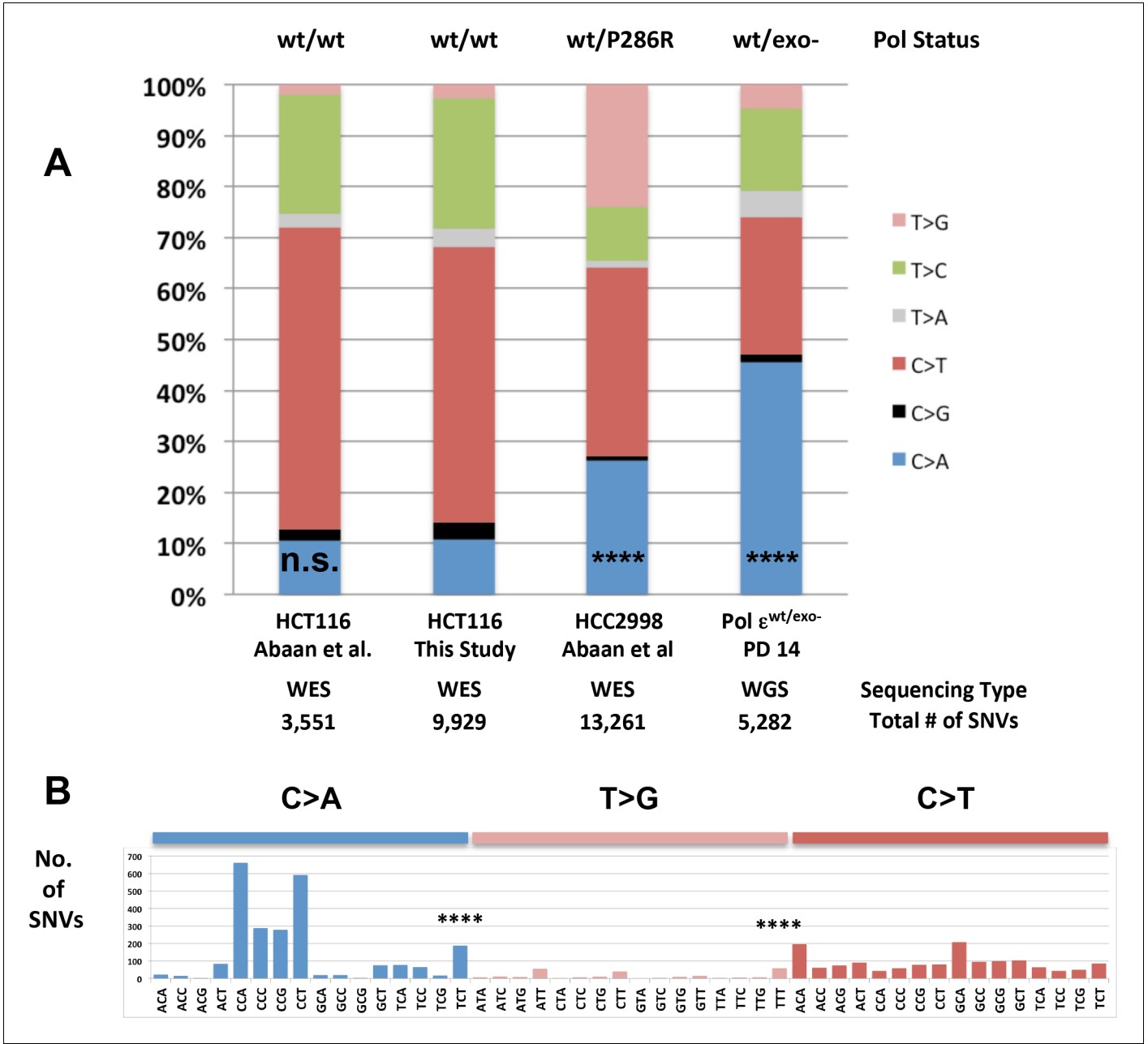

**Figure 2.** Whole-genome sequencing from defined population doubling Pol ε[wt/exo-] mismatch repair-deficient cells. (A) Whole genome sequencing (2.8 × 10⁹ bp, average 30X coverage) was performed on Pol ε[wt/exo-] cells lacking mismatch repair at two defined population doubling levels, P0 and P14, as described in the Methods. P0 was used as the matched normal cells to define only those mutations arising during the 14 population doublings. The fraction of each type of base pair substitution from the PD 14 Pol ε[wt/exo-] cells was plotted and compared to the fraction of each type of mutation from HCT116 ((*Abaan et al., 2013*) and this study) and HCC2998 cells (*Abaan et al., 2013*). Chi square tests with Yates correction were used to calculate p values relative to SNVs found in Pol ε[wt/wt] mismatch repair-deficient cells in this study. Pol ε[wt/wt] (Abaan et al.) $\chi^2$ = 0.033, p=0.8551; Pol ε[wt/P286R] (Abaan et al.) $\chi^2$ = 872.341, p<0.0001; Pol ε[wt/exo-] $\chi^2$ = 2,3680.508, p<0.0001. ****p<0.0001; n.s., not significant. (B) The number of each indicated base pair substitution in a specific trinucleotide context was plotted from the PD 14 Pol ε[wt/exo-] mismatch repair-deficient cells. The base pair substitutions shown (C > A and T > G transversions, left; C > T transitions, right) are those found enriched in POLE tumors. Chi square tests with Yates correction were used to calculate p-values relative to SNVs found in Pol ε[wt/wt] mismatch repair-deficient cells in this study. C > A TCT $\chi^2$ = 152.772, p<0.0001; T > G TTT $\chi^2$ = 72.254, p<0.0001. ****p<0.0001.

DOI: https://doi.org/10.7554/eLife.32692.008

The following figure supplements are available for figure 2:

**Figure supplement 1.** Whole genome SNVs identified in Pol ε[wt/exo-] (PDL = 14) in cells lacking functional mismatch repair identified.

DOI: https://doi.org/10.7554/eLife.32692.009

*Figure 2 continued on next page*

*Figure 2 continued*

**Figure supplement 2.** POLE mutation signature extracted from POLE-mutant cell lines.

DOI: https://doi.org/10.7554/eLife.32692.010

**Figure supplement 3.** Mutation counts in the indicated trinucleotide context (*top*) were plotted as a proportion of their occurrence (*bottom*) in WGS samples.

DOI: https://doi.org/10.7554/eLife.32692.011

**Figure supplement 4.** Mutation counts in the indicated trinucleotide context (*top*) were plotted as a proportion of their occurrence (*bottom*) in WES samples.

DOI: https://doi.org/10.7554/eLife.32692.012

**Figure supplement 5.** Relative contributions of Cosmic Mutation Signatures to individual patient mutation spectra were determined using deconstructSig.

DOI: https://doi.org/10.7554/eLife.32692.013

**Figure supplement 6.** Mean coverage was greater than 90x for each WES sample.

DOI: https://doi.org/10.7554/eLife.32692.014

**Figure supplement 7.** Alignment rate to the reference genome exceeded 99% for each WES sample.

DOI: https://doi.org/10.7554/eLife.32692.015

**Figure supplement 8.** Total reads exceeded 60 million for each WES sample.

DOI: https://doi.org/10.7554/eLife.32692.016

**Figure supplement 9.** Greater than 90% of the bases in the WES genome exceeded 30x coverage.

DOI: https://doi.org/10.7554/eLife.32692.017

**Figure supplement 10.** Greater than 85% of the bases in the WGS genome exceeded 20x coverage.

DOI: https://doi.org/10.7554/eLife.32692.018

**Figure supplement 11.** Average alternate base quality to reference base quality of ~1.0.

DOI: https://doi.org/10.7554/eLife.32692.019

significantly higher rate in vitro than the wild type exonuclease-proficient Pol ε enzyme (*Figure 1C*). We used a construct comprised of the N-terminal 140 kDa of Pol ε, which contains the DNA polymerase and exonuclease domains and has similar fidelity and catalytic activity to the complete four subunit holoenzyme (*Aksenova et al., 2010*; *Ganai et al., 2015*; *Zahurancik et al., 2015*). Importantly, the elevated TCT→TAT error rate we observed with the D275A/E277A mutant was not statistically different from those measured with the S459F and S461P Pol ε cancer mutants previously (*Shinbrot et al., 2014*; *Shlien et al., 2015*), suggesting a common mechanism of mutagenesis for these hotspot mutations.

Mutation rates calculated using reporter genes ($\mu_L$) can be used to extrapolate to genome-wide per base pair mutation rates ($\mu_{BS}$) (*Drake, 1991*; *Lynch, 2010*). The availability of high-throughput DNA sequencing now allows for empirical validation of these calculations in addition to providing insight into the influence of genomic context on mutagenesis. To address this we performed whole-genome sequencing ($2.8 \times 10^9$ bp at an average depth of 36.1x) on genomic DNA prepared from Pol ε$^{wt/exo-}$ cells. Based on our measured mutation rate for HPRT1 ($\mu_L$) in Pol ε$^{wt/exo-}$ cells lacking mismatch repair ($180 \times 10^{-7}$), we calculated a $\mu_{BS}$ value of $0.23 \times 10^{-7}$ mutations per base pair per genome duplication.

Because the parental HCT-116 cell line already carries a significant number of single nucleotide variants (SNVs) relative to the human reference sequence ([*Abaan et al., 2013*] and see Discussion), we needed a way of measuring de novo mutations resulting from Pol ε-dependent replication errors. To do this we first performed whole genome sequencing (WGS) on genomic DNA prepared from mismatch repair-deficient Pol ε$^{wt/exo-}$ cells, which we then used as a matched normal control. We termed this mutation spectrum P0. We then passaged these cells through a calculated 13.9 population doublings and then performed WGS again on the passaged population, which we termed P14. Mutations unique to P14 arose during the defined number of population doublings. The P0 and P14 samples contained 140.3 and 141.4 Mut/Mb, respectively. Given the calculated $\mu_{BS}$ and the $2.8 \times 10^9$ bp sequenced, we predicted the accumulation of 906 novel genome-wide mutations after 14 population doublings. Whole-genome sequencing revealed 5,282 SNVs unique to the P14 population, 5.8-fold higher than that predicted from the $\mu_L$ at HPRT1. Mutations observed in HPRT1 in this cell line may thus slightly underrepresent those found genome-wide. This difference is consistent with what is seen in microbes, where reporter gene mutation rates are consistently 6–8-fold lower than concurrently measured whole-genome mutation rates, likely due to phenotypic lag, strong

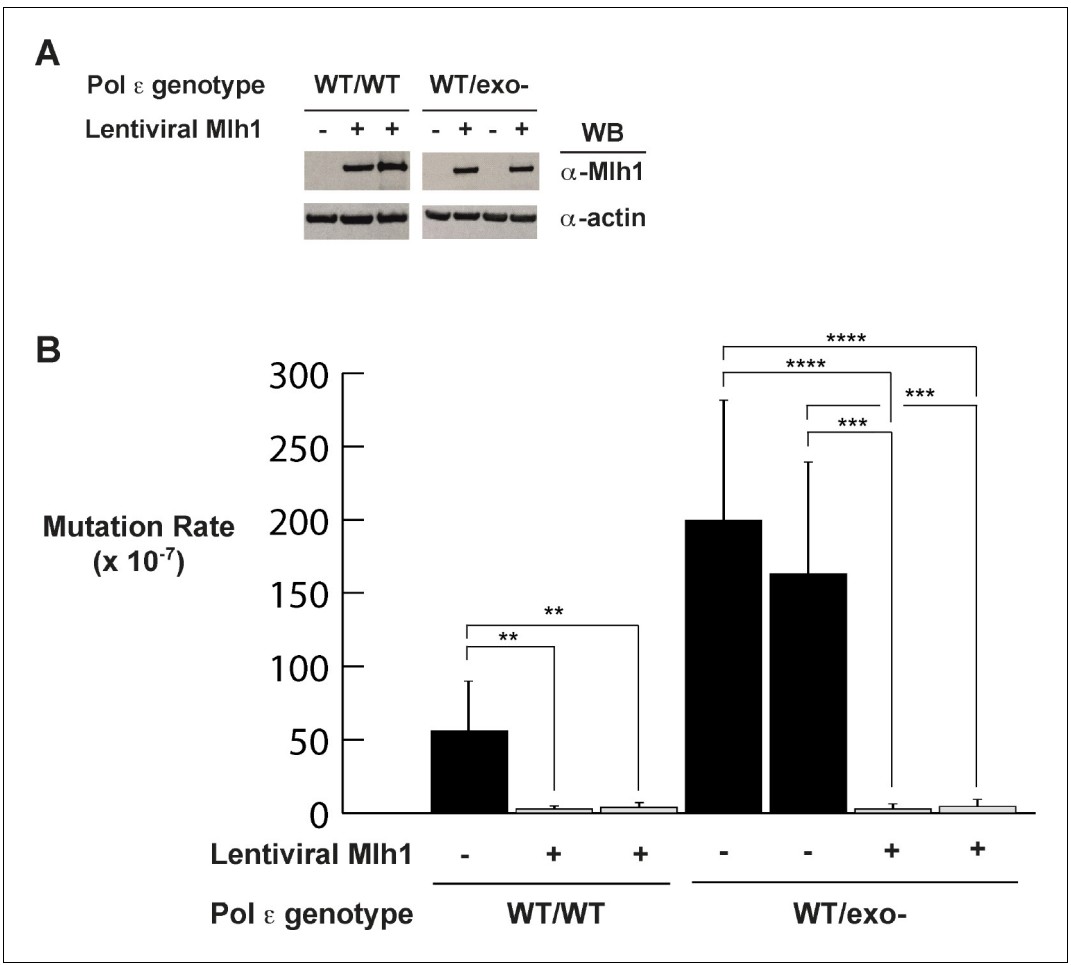

**Figure 3.** Mismatch repair suppresses exonuclease-deficient Pol ε-induced mutation rate increase. (**A**) Lentivirus encoding human Mlh1 was generated and used to infect parental cells with wild type Pol ε and cells heterozygous for Pol ε exonuclease deficiency. Cell lysates were probed by Western blot using antibodies against Mlh1 and β-actin. (**B**) Mutation rates were measured by fluctuation analysis as described in the Methods using the Ma-Sandri-Sarkar Maximum Likelihood Estimator. Twelve independent isolates from each of two parental (wt/wt) and two heterozygous cell lines (wt/exo-) expressing Mlh1 were used. 95% confidence intervals are shown. Pol $ε^{wt/wt}$ Mlh1+ Clone 1 Mutation Rate = $1.7 \times 10^{-7}$, SEM = $0.72 \times 10^{-7}$, p=0. 0046. Pol $ε^{wt/wt}$ Mlh1+ Clone 2 Mutation Rate = $2.5 \times 10^{-7}$, SEM = $1.1 \times 10^{-7}$, p=0.0053. Pol $ε^{wt/exo-}$ Mlh1+ Clone 1 Mutation Rate = $2.3 \times 10^{-7}$, SEM = $0.81 \times 10^{-7}$, p<0.0001 (vs. Pol $ε^{wt/exo-}$ Mlh1- Clone 1) and p=0.0003 (vs. Pol $ε^{wt/exo-}$ Mlh1- Clone 2). Pol $ε^{wt/exo-}$ Mlh1+ Clone 2 Mutation Rate = $3 \times 10^{-7}$, SEM = $1.3 \times 10^{-7}$, p<0.0001 (vs. Pol $ε^{wt/exo-}$ Mlh1- Clone 1) and p=0.0003 (vs. Pol $ε^{wt/exo-}$ Mlh1- Clone 2). Mutation Rates for Pol $ε^{wt/exo-}$ Mlh1+ Clone 1 and Clone 2 were not significantly different (p=0.6485). Mutation rates from cells lacking mismatch repair (from *Figure 1A*) are shown for comparison.
DOI: https://doi.org/10.7554/eLife.32692.020

---

selective pressure and transcription in the reporter gene (*Drake, 2012*; *Jee et al., 2016*; *Lee et al., 2012*).

C→A transversions exceeding 20% of all base pair substitutions is a primary characteristic of mutation spectra from tumors containing Pol ε exonuclease domain mutations (*Rayner et al., 2016*; *Shinbrot et al., 2014*). C→A transversions were increased significantly in the Pol $ε^{wt/exo-}$ cells as compared to the control Pol $ε^{wt/wt}$ spectrum, accounting for 46% of all base pair substitutions (*Figure 2A*, $χ^2$ = 11.874, p<0.0001). These were not cell line artifacts, as whole exome sequencing from HCT-116 cells from two independent studies ([*Abaan et al., 2013*] and this study) showed roughly 10% C→A transversions (*Figure 2A*, p>0.5). HCC2998 cells, which harbor the Pol $ε^{wt/P286R}$ mutation, also showed a significant increase in C→A transversions relative to Pol $ε^{wt/wt}$ cells (*Figure 2A*, p<0.0001).

Two sequence context mutational hotspots were observed that are consistent with Pol ε exonuclease domain mutant spectra: C→A transversions in TCT context and T→G transversions in TTT context and, to a lesser extent, ATT and GTT contexts (*Figure 2B*). These hotspots are seen in Pol ε tumors from patients with bMMRD (*Shlien et al., 2015*), colorectal and endometrial cancer (*Alexandrov and Stratton, 2014*; *Cancer Genome Atlas Network, 2012*; *Kandoth et al., 2013*; *Shinbrot et al., 2014*), as well as in the Pol ε-P286R HCC2998 cells (*Figure 2—figure supplement 1*, data extracted from [*Abaan et al., 2013*]). These are not mutational hotspots in HCT-116 cells, which contain wild type Pol ε (*Figure 2—figure supplement 1*). The largest number of mutations that arose during the 14 doublings were C→A transversions in triplet contexts containing adjacent cytosines: CCA, CCT, CCC and CCG. Triplet nucleotide occurrences can vary in the regions captured by WGS and WES. In order to address this we reanalyzed each sample relative to the number of times each trinucleotide is found in the relevant sample and found the hotspot patterns are all retained (*Figure 2—figure supplements 3–4*). The increase in C→A mutations in the CCT context was also seen in Pol ε exonuclease domain (EDM) tumors from bMMRD patients (*Shlien et al., 2015*), suggesting a link between Pol ε replication errors left uncorrected by mismatch repair. C→A mutations in CCA, CCC and CCG contexts are slightly elevated in Mutation Signature 20, which has been associated with loss of mismatch repair (*Alexandrov et al., 2013b*). These transversions were seen in the HCT116 cell line with wild type Pol ε (*Figure 2—figure supplement 1*), though to a lesser extent. The lack of C→T transitions in TCG contexts is significantly different from colorectal and endometrial Pol ε tumors, but consistent with their absence from bMMRD tumors with Pol ε EDM mutations (*Cancer Genome Atlas Network, 2012*; *Kandoth et al., 2013*; *Shinbrot et al., 2014*).

## Expression of MMR suppresses Pol ε<sup>wt/exo-</sup> mutagenesis

While it is clear that Pol ε-dependent mutagenesis in the absence of functional MMR accounts for the ultramutated phenotype in bMMRD tumors with Pol ε mutations, the role of MMR in Pol ε somatic tumors is less clear. In order to measure the effects of MMR on Pol ε exonuclease-dependent replication errors, we wanted to measure error rates in both the presence and absence of MMR. Previous studies have restored MMR by stably adding Mlh1-expressing chromosome 3 to cells (*Glaab and Tindall, 1997*). We made Mlh1-encoding lentivirus and used this to infect Mlh1-deficient HCT-116 cells containing wild type and mutant Pol ε (*Figure 3A*). Lentiviral Mlh1 expression reduced mutation rates at the HPRT1 locus by 14- to 20-fold in the wild type polymerase background (*Figure 3B*), similar to the 12-fold reduction reported when the Mlh1-encoding chromosome 3 was added back to HCT-116 cells ([*Glaab and Tindall, 1997*; *Tindall et al., 1998*]; $73 \times 10^{-7}$ and $5.9 \times 10^{-7}$; 12.4-fold reduction), indicating that the expressed Mlh1 is functional.

Mlh1 expression in Pol ε<sup>wt/exo-</sup> cells caused an over 50-fold decrease in the mutation rate (to 2.3 and $3.0 \times 10^{-7}$, *Figure 3B*), making them indistinguishable from those measured in Pol ε<sup>wt/wt</sup> cells with Mlh1 expressed (*Figure 3B*). This result also suggests that Msh3 is unlikely to play a significant role in correcting the exonuclease-deficient Pol ε errors since HCT-116 cells are deficient in this factor and it was not added back in these experiments (*Papadopoulos et al., 1994*).

When fluctuation assay mutation rates are very low due to a significant number of independent isolates giving rise to zero HPRT1-mutant colonies, as was the case here, an alternative method to measure mutation rates can be used. We chose to periodically measure HPRT1 mutant frequencies at increasing population doubling level (PDL), where the slope of the plotted line is equal to the mutation rate (*Glaab and Tindall, 1997*). We measured HPRT1 mutant frequencies at several population doublings from PDL = 0 to PDL = 70 or 71 in Pol ε<sup>wt/wt</sup> and Pol ε<sup>wt/exo-</sup> cells expressing Mlh1, respectively (*Figure 4A*). At each PDL we scored between 1 and 19 6-TG-resistant colonies. However, when we sequenced the HPRT1 ORF from all 6-TG-resistant colonies we saw many instances of repeat mutations in a collection from a single PDL, indicative of a single mutational event that expanded throughout the population. Plotting mutant frequency values calculated for the indicated PDL using only the unique HPRT1 mutations (*Figure 4—source data 1*) returned a line with slope of ~1, suggesting that the mutation rates were at or near the level of detection of this assay. The Pol ε<sup>wt/exo-</sup> mutant frequencies were consistently higher than those from the Pol ε<sup>wt/wt</sup> cells, but this difference was not statistically significant (*Figure 4A*).

To determine if this phenomenon held throughout the genome, we carried out whole-exome sequencing to an average depth of 100x on the early (PDL = 0) and late (PDL = 70) samples from

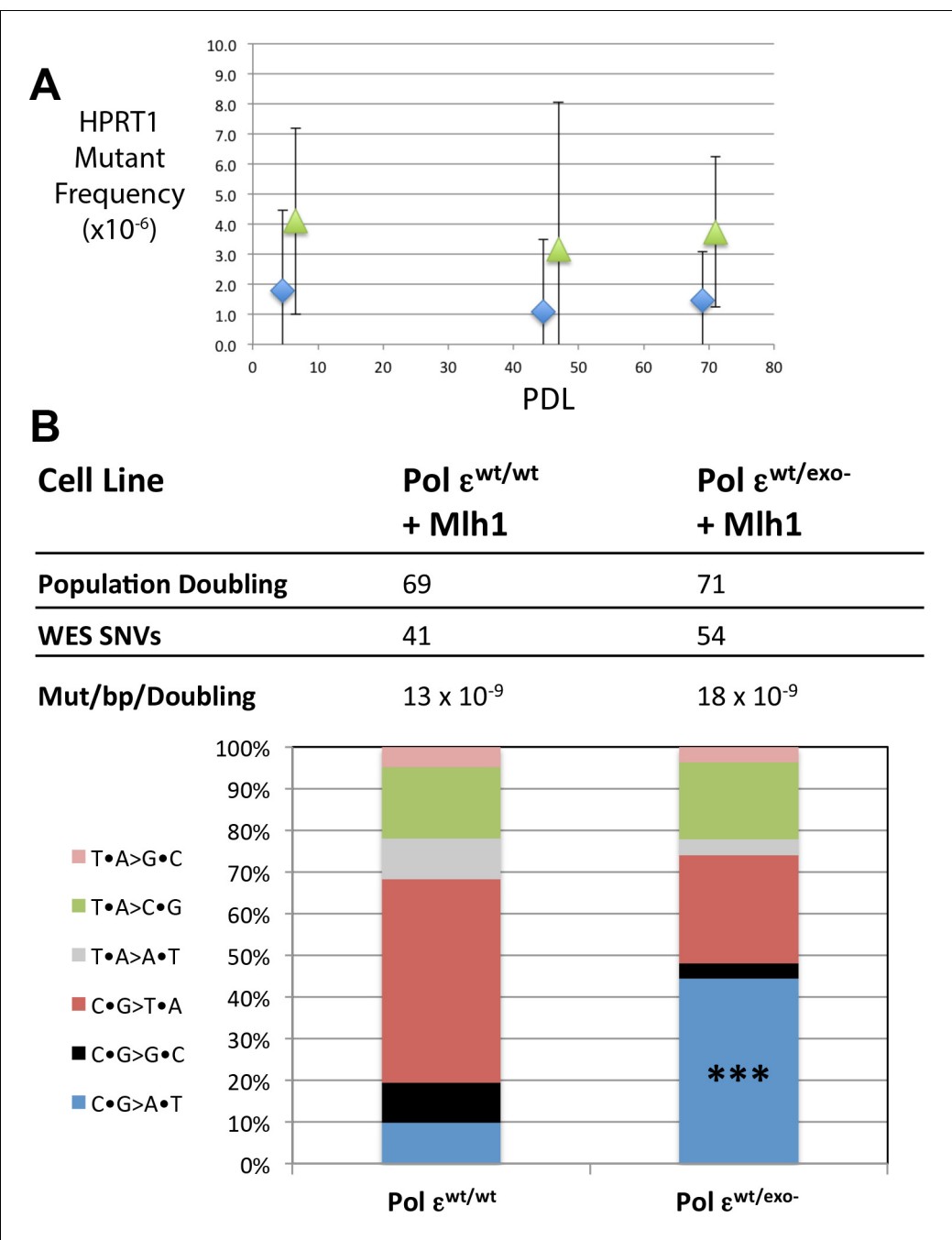

**Figure 4.** Mismatch repair suppresses the majority of exonuclease-deficient Pol ε mutation specificity. (**A**) Cells were continuously passaged and PDL was calculated using the following equation: PDL = $[\ln(N_t)-\ln(N_0 \cdot PE)]/\ln2$. $N_t$ = Number of viable cells counted after passage; $N_0$ = Number of cells seeded prior to passage; PE = plating efficiency. Mutant frequencies were measured for each mismatch repair proficient strain at the indicated PDL (diamonds, Pol ε$^{wt/wt}$; triangles, Pol ε$^{wt/exo-}$). Ten plates for each cell lines were seeded with $2 \times 10^5$ cells at each PDL into media containing 6-TG and grown for 12–14 days. Each 6$^{TG}$-resistant clone was isolated, expanded and the HPRT1 ORF was sequenced. Mutant frequencies were calculated based on the number of unique HPRT1 mutations at each PDL. Pol ε$^{wt/wt}$ PDL6.4 MF = $1.8 \times 10^{-6}$, SEM = $2.7 \times 10^{-6}$, n = 4; Pol ε$^{wt/exo-}$ PDL6.6 MF = $4.1 \times 10^{-6}$, SEM = $3.1 \times 10^{-6}$, n = 3, p=0.6003. Pol ε$^{wt/wt}$ PDL44.6 MF = $1.1 \times 10^{-6}$, SEM = $2.3 \times 10^{-6}$, n = 2; Pol ε$^{wt/exo-}$ PDL47.9 MF = $3.2 \times 10^{-6}$, SEM = $4.7 \times 10^{-6}$, n = 8, p=0.9066. Pol ε$^{wt/wt}$ PDL69 MF = $1.5 \times 10^{-6}$, SEM = $1.6 \times 10^{-6}$, n = 5; Pol ε$^{wt/exo-}$ PDL71 MF = $3.7 \times 10^{-6}$, SEM = $2.6 \times 10^{-6}$, n = 5, p=0.4917. (**B**) Whole exome sequencing ($30 \times 10^6$ bp, average 101x coverage) was performed on the indicated cell line at two defined population doubling levels, P0 and P69 or P71, as described in the Methods. P0 for each cell line was used as the matched

*Figure 4 continued on next page*

*Figure 4 continued*

normal cells to define only those mutations arising during the 70 or 71 population doublings. The fraction of each type of base pair substitution found unique to PDL 69 (for Pol ε$^{wt/wt}$) or PDL 71 (Pol ε$^{wt/exo-}$) was plotted and compared. Fisher's exact tests were used to calculate p values. p=0.0002 (***p<0.001).

DOI: https://doi.org/10.7554/eLife.32692.021

The following source data and figure supplement are available for figure 4:

**Source data 1.** HPRT1 mutations sequenced from mismatch repair-proficient cells.

DOI: https://doi.org/10.7554/eLife.32692.023

**Source data 2.** Pol ε mutation spectra calculation of cosine similarity to cancer mutation spectra.

DOI: https://doi.org/10.7554/eLife.32692.024

**Figure supplement 1.** Whole exome SNVs identified in Pol ε$^{wt/exo-}$ (PDL = 71) and Pol ε$^{wt/wt}$ (PDL = 70) cells expressing Mlh1 were analyzed for their triplet nucleotide sequence context.

DOI: https://doi.org/10.7554/eLife.32692.022

both Pol ε$^{wt/wt}$ and Pol ε$^{wt/exo-}$ mismatch repair-proficient cell lines (*Figure 4B*). Using the PDL = 0 samples as matched normal controls, we measured similar low mutation rates in Pol ε$^{wt/wt}$ and Pol ε$^{wt/exo-}$ cells (13 × 10$^{-9}$ Mut/bp/doubling and 18 × 10$^{-9}$ Mut/bp/doubling, respectively). The total numbers of all mutations acquired were essentially no different than with wild type Pol ε. Interestingly, there was a statistically significant increase in C→A transversions (p=0.0002) between the mismatch repair-proficient Pol ε$^{wt/exo-}$ cells and the mismatch repair-proficient Pol ε$^{wt/wt}$ cells, while no statistically significant difference was found in any other class of base pair substitution (p>0.2 for each of the six classes, Fisher's Exact Test). Further, all triplet context mutations were observed in insufficient numbers to evaluate statistically. C→A mutations were, however, observed in all triplet contexts seen as hotspots in the MMR-deficient cells (CCA, CCC, CCG, CCT and TCT, *Figure 4—figure supplement 1*). Mutation signature 10, the unique Pol ε mutation signature, was extracted from Pol ε exonuclease-deficient mutation spectra from cells with and without mismatch repair (*Figure 2—figure supplement 2* and *Figure 4—source data 2*). The relative contribution of signature 10 in Pol ε exo-deficient cells is closer to that seen in bMMRD patients (*Figure 2—figure supplement 5*), most likely due to the relative absence of C→T transitions in TCG context. These results indicate that the majority of replication errors made by the Pol ε-D275A/E277A mutant are in fact corrected by mismatch repair.

## Discussion

In the current study we examined the relative contributions of two essential determinants of replication fidelity, proofreading and mismatch repair, on mutagenesis in human cells. We used a combination of gene editing, reporter gene studies and next generation sequencing to measure mutation rates and specificities in human cells engineered to model proofreading-deficient tumors with and without mismatch repair. This is the equivalent to what occurs in human tumors with mutations in the Pol ε exonuclease domain and genomic mutation frequencies exceeding 100 mutations per Mb (*Cancer Genome Atlas Network, 2012*; *Rayner et al., 2016*; *Shinbrot et al., 2014*; *Shlien et al., 2015*). We show that large and rapid mutation accumulation occurs when Pol ε exonuclease domain mutations occur along with inactivation of mismatch repair. Most of these are specific transversion mutations known to be hotspots of exonuclease-deficient Pol ε mutagenesis. We further show that this large increase in mutation rate is largely suppressed by functional mismatch repair. Taken together, these results suggest that the mechanism of replication error mutagenesis in sporadic tumors with heterozygous Pol ε mutations likely requires an additional feature, several of which are described below, including suppression of MMR and alternative effects on Pol ε activity.

We used rAAV-mediated gene targeting to replace two exonuclease active site residues, D275 and E277, with alanines on a single POLE allele. The single allelic inactivation was chosen to model the case in tumors with heterozygous Pol ε mutations. This double amino acid substitution has been shown to inactivate exonuclease proofreading in vitro and cause increased reporter gene mutation rates in yeast and mammalian cells (*Morrison et al., 1991*; *Morrison and Sugino, 1994*; *Tran et al., 1999*; *Albertson et al., 2009*; *Korona et al., 2011*; *Shcherbakova and Pavlov, 1996*; *Agbor et al., 2013*). Next generation sequencing on these cells in the presence or absence of mismatch repair over defined

numbers of population doublings allowed us to compare genome-wide mutation rates and spectra to the mutation spectra from patient tumors.

Unbiased whole-genome sequencing confirmed the rapid accumulation of Pol ε-specific mutations seen in POLE tumors lacking functional mismatch repair (*Shlien et al., 2015*). The total number of measured SNVs suggests a mutation rate of 380 mutations per population doubling, similar to the 608 mutations per cell cycle calculated for a mismatch repair-deficient brain tumor harboring a Pol ε exonuclease domain mutation. Our cellular mutation rate values possibly underestimate the true Pol ε exonuclease-deficient mutation rate for several reasons. Our data were generated from a cancer cell line with a large number of pre-existing mutations (*Abaan et al., 2013*), as well as additional mutations that have assuredly arisen during passaging in the laboratory. These could conceivably include suppressor mutations functioning to restrain elevated mutation rates (*Morrison and Sugino, 1994*; *Herr et al., 2011a*; *Williams et al., 2013*). Importantly, no additional mutations in POLE were sequenced, suggesting that viability of this cell line is not due to an acquired mutation elsewhere in POLE acting to suppress the mutation rate, as occurs frequently in yeast (*Herr et al., 2011a*; *Williams et al., 2013*; *Herr et al., 2011b*; *Dennis et al., 2017*). While we cannot formally exclude the possibility that a de novo mutation in another gene acted to suppress the mutation rate in trans, no obvious candidates were identified.

An additional reason that our mutation rates may underestimate the true mutation rate is that mutations that arise in the last several rounds of replication and those that fall below 5% allele frequency would not meet the threshold for scoring as a true SNV. The genome data was generated using an instrument with high accuracy (<1% error rate) and variants were called using an established algorithm, however there are indeed a small number of areas in the genome that are inaccessible – either due to gaps in the reference assembly, or excessive numbers of repeats that prevent proper alignment. Experiments using single-cell sequencing could address these issues, ideally by selecting single cells, expanding subclones and then measuring mutations at higher stringency values than used here. These rates are also similar to the per base pair mutation rates in haploid yeast with complete Pol ε exonuclease deficiency and disrupted MMR (*Kennedy et al., 2015*). This similarity is striking considering our measurements were made in a heterozygous diploid human cell line. A key finding from the yeast study was that individual cell mutation rates could vary by an order of magnitude. We are currently unable to measure mutation rates in individual cells, but this remains a critical issue to address in future studies.

The unique mutation spectrum seen in POLE tumors was recapitulated in our gene-targeted cell lines, with one notable exception. In tumors, many C→A transversions occur in a highly specific triplet sequence context, TCT, which we also see in the cell lines, though not to the same proportion as in the tumor genomes. Interestingly, this particular mutation is also enriched in yeast with the P286R equivalent allele (*Barbari and Shcherbakova, 2017*). We also observe increased T→G transversions in TTT (and to a lesser extent ATT and CTT) context, similar to Pol ε tumors. Because of the limited number of mutational target sites we cannot at this time draw conclusions as to Pol ε strand usage during replication. Experiments designed to assess strand bias in these errors are currently underway. What is notable, however, is the lack of TCG→TTG transitions in our dataset. This is the second most frequent Pol ε-specific mutation in the TCGA database. TCG→TTG transitions were also not found elevated in the Pol ε bMMRD brain tumor mutation spectrum. This difference may reflect interesting, but as-yet undefined tissue differences.

Another possible explanation for these differences is that the Pol ε mutants found in tumors are somehow intrinsically different biochemically from the double alanine substitution mutant used in the current study. Depending on the reporter gene used, the monoallelic Pol ε-P286R mutant is a 2.3- to 12-fold stronger mutator than the pol2-4 mutant (equivalent to the human Pol ε D275A/E277A studied here) when measured in a diploid yeast strain (*Kane and Shcherbakova, 2014*). However, a number of direct biochemical comparisons of activity and fidelity (*Figure 1C*, (*Shinbrot et al., 2014*; *Shlien et al., 2015*) and unpublished observations) between several cancer mutant constructs and the D275A/E277A construct have not yet shown any significant differences that could account for this. Certain DNA Pol mutants, including some found in human tumors, can cause increased mutagenesis by inducing expansions of normal dNTP pools in yeast and human cells (*Dennis et al., 2017*; *Mertz et al., 2015*; *Williams et al., 2015*). Interestingly, the pol2-4 allele has no effect on dNTP pools in yeast, suggesting a possible explanation for possible allelic differences with functional MMR.

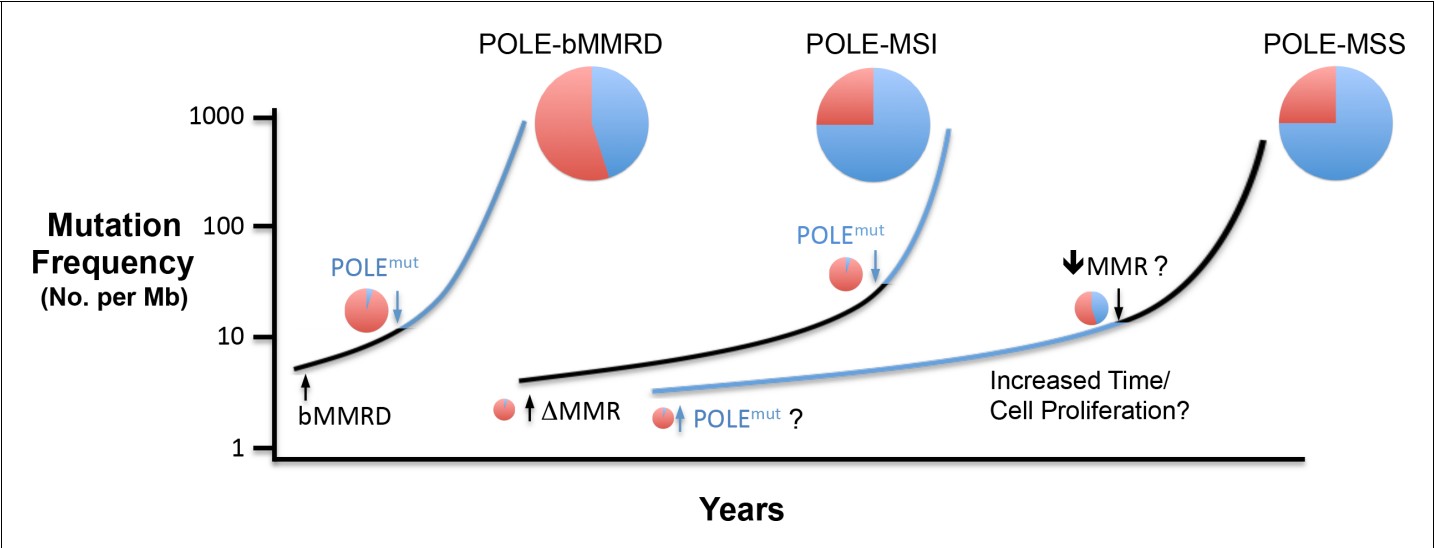

**Figure 5.** Model for Pol ε-dependent tumor mutation signature development. Rapid, massive mutation accumulation and Pol ε mutation signature acquisition (blue circles) depends on both Pol ε exonuclease domain mutation and compromised mismatch repair function. In somatic tumors, the partial MSI phenotype seen in a subset of POLE patients is likely the result of mismatch repair loss preceding Pol ε mutation (black line), leading to an accumulation of Pol ε-independent mutations (red circles). Mutations in bMMRD patients develop with similar mutation patterns, but accelerated timing due to germline loss of mismatch repair. When the Pol ε mutation occurs first during somatic tumor development, the mutation signature likely requires an additional characteristic for the explosive mutation acquisition to occur (blue line). Possibilities include subsequent suppression of mismatch repair (↓MMR?), unique biochemical properties (POLE^mut?) or increased time and or cellular proliferation.

DOI: https://doi.org/10.7554/eLife.32692.025

The following figure supplement is available for figure 5:

**Figure supplement 1.** Oncoprints were made using cBioPortal for colorectal (n = 8) and endometrial (n = 18) tumors from the TCGA studies containing Pol ε exonuclease domain mutations.

DOI: https://doi.org/10.7554/eLife.32692.026

In heterozygous Pol ε^wt/exo- cells with functional mismatch repair, mutation rates were suppressed to the levels seen in cells with wild type Pol ε. These rates would be insufficient to give rise to ultra-hypermutated tumors in a matter of months. In addition, there is no explosive accumulation of triplet context-specific mutations in the MMR-proficient Pol ε^wt/exo- cells that is seen in these tumors.

Given that the HCT-116 cells used in these studies are mutators themselves, it is possible that pre-existing deficiencies in other DNA repair or replication proteins could contribute to the observed mutagenesis. While direct contribution is unlikely given the absence of POLE mutation spectrum in the wild type Pol ε cells, cooperation with exonuclease-deficient Pol ε remains a formal possibility. To address this we used gene ontology to identify 58 DNA repair and replication proteins mutated in our HCT-116 cells, including 38 non-synonymous and 20 indel mutations. While several interesting candidates with known links to mutagenesis were identified, all have been shown by other groups to be expressed in this cell line and each, when tested, is functional (e.g. ATM, SETD2, Pol η, Pol ζ [*Bhat et al., 2013*; *Hahn et al., 2011*; *Nicolay et al., 2012*; *Zhou et al., 2013*; *Zhu et al., 2009*]). No mutation that arose during the population doubling experiments showed any obvious link to mutagenesis.

Our results support a model in which simple heterozygous loss of two Pol ε exonuclease metal chelating residues on a single allele of POLE is insufficient to drive Pol ε ultramutational specificity. Additional factors are likely required to help drive the ultramutated phenotype observed in POLE tumors, including suppression of mismatch repair, discussed below. In bMMRD, the complete lack of mismatch repair prior to Pol ε mutation leads to the moderate accumulation of Pol ε-independent replication errors (*Figure 5*). Mutation rates then increase dramatically upon loss of proofreading in one allele, with the Pol ε error signature representing a smaller fraction of the total errors, which is seen in these tumors (*Shlien et al., 2015*). Our results suggest that Pol ε mutations in somatic tumors can occur first and early, but later suppression of MMR would then accelerate overall mutation rates

to that seen in the ultramutated tumors, while the signature mutation proportion remains high (*Cancer Genome Atlas Network, 2012*; *Kandoth et al., 2013*).

Analysis of the mutational status of all mismatch repair genes in Pol ε tumors sequenced by TCGA supports the model of mismatch repair loss dramatically accelerating the acquisition of Pol ε-specific mutations. 85% (22/26) of the TCGA Pol ε tumors also have a mutation in at least one mismatch repair gene, most of which (18/22) harbor at least one nonsense mutation, which are more likely to be inactivating mutations (*Figure 5—figure supplement 1*). This predicts that at least some tumors would show evidence of MSI. In the original TCGA studies, several POLE tumors were actually first classified as MSI (three as MSI-H; five as MSI-L) (*Cancer Genome Atlas Network, 2012*; *Kandoth et al., 2013*). Analysis of sequencing reads from 46 homonucleotide runs in the POLE endometrial tumors showed no evidence of instability, so the POLE tumors were then reclassified as MSS (*Shinbrot et al., 2014*). However, the initial TCGA studies used both homo- and di-nucleotide loci to score MSI, raising the possibility that a subset of POLE tumors have a microsatellite instability defect at repeats more complex than homonucleotides. Indeed, the repeat unit size, the number of repeats and the repeat sequence composition are known to have very strong influences on the variability of microsatellite mutagenesis (*Shah et al., 2010*). Curiously, however, most (15/18) of the MMR gene nonsense mutations are the result of TCT→TAT transversions, raising the possibility that Pol ε mutation occurs first and possibly even promotes subsequent mutational inactivation of MMR.

Of all the Pol ε mutant colorectal and endometrial tumors sequenced in the TCGA studies, 15% (4/26) lacked a mutation in any mismatch repair gene and also showed no evidence of MLH1 promoter hypermethylation, demonstrating that the ultramutated phenotype can arise when mismatch repair is intact at both the genetic and epigenetic level. An alternative possibility is that mismatch repair activity is suppressed at some point during POLE tumor development. In this scenario, mutations introduced by the mutant Pol ε could accumulate slowly even in the presence of genotypically and epigenetically wild type mismatch repair. A number of conditions have been shown to transiently and reversibly lower mismatch repair protein levels and inhibit mismatch repair activity, including hypoxia, oxidative damage, inflammation, reduced pH, exposure to adriamycin or cadmium and treatment with mutagenic dNTP analogs (*Banerjee and Flores-Rozas, 2005*; *Francia et al., 2005*; *Larson and Drummond, 2001*; *Mihaylova et al., 2003*; *Chang et al., 2002*; *Hile et al., 2013*; *Iwaizumi et al., 2013*; *Lu et al., 2014*; *Negishi et al., 2002*). The variable nature and duration of such a suppression event would be expected to result in a complex effect on microsatellite instability. Perhaps even more intriguingly, transient mismatch repair suppression has been seen in the context of proofreading-deficiency in E. coli (*Fijalkowska and Schaaper, 1996*; *Schaaper and Radman, 1989*). While replication errors made by the proofreading-deficient allele tested here were clearly insufficient to suppress MMR, it is possible that the nature and rate of errors made by cancer-associated alleles might be sufficient to saturate and overwhelm MMR pathways.

Our results support the idea that loss of a single Pol ε proofreading allele is sufficient to drive a subset of the observed clinical characteristics of Pol ε tumors, provided mismatch repair is compromised in some way. These observations further support the idea that in the presence of fully functional MMR the appearance of the ultrahypermutated mutation signature may be more directly related to some as yet uncharacterized additional defect in the mutant polymerase (*Barbari and Shcherbakova, 2017*). These ideas are not mutually exclusive of one another.

Given the recent success of immune checkpoint therapies in treating tumors with high mutation burden (*Shlien et al., 2015*; *Bouffet et al., 2016*; *Hodi et al., 2010*; *Le et al., 2015*; *Santin et al., 2016*), it is of great interest to understand the mechanisms that result in ultrahypermutated tumors harboring DNA polymerase mutations.

## Materials and methods

**Key resources table**

| Reagent type (species) or resource | Designation | Source or reference | Identifiers | Additional information |
|---|---|---|---|---|
| cell line (Homo sapiens, Male) | HCT116 cells | Other | RRID:CVCL_0291 | Prescott Deininger at Tulane Univeristy LCRC |

*Continued on next page*

*Continued*

| Reagent type (species) or resource | Designation | Source or reference | Identifiers | Additional information |
|---|---|---|---|---|
| cell line (H. sapiens, Male) | HCT116 + Mlh1 | This paper | NA | HCT116 cells stably expressing human Mlh1-ORF via lentivirus-mediated integration |
| cell line (H. sapiens, Male) | Exo-; Exonuclease-deficient HCT116 Cells | This paper | NA | HCT116 cells infected with rAAV containing D275A and E277A POLE mutations |
| cell line (H. sapiens, Male) | Exo-; Exonuclease-deficient HCT116 Cells + Mlh1 | This paper | NA | HCT116 cells stably expressing human Mlh1-ORF via lentivirus-mediated integration and infected with rAAV containing D275A and E277A POLE mutations |
| recombinant DNA reagent | ExoI-targeting rAAV vector | This paper | NA | Homology arms/SEPT Cassette/Exo- mutations |
| recombinant DNA reagent | pCMV-XL5-Mlh1 | Other | NA | Victoria Belancio at Tulane Univeristy LCRC |
| antibody | Mlh1 Antibody | Pharmingen | G168-728; RRID: AB_395227 | Rabbit monoclonal; (1:100) in Milk (1%) TBST (1X) x 1 hr at RT |
| chemical compound, drug | 6-Thioguanine; 6-TG | Sigma-Aldrich | A4882 | Used at 5 ug/mL final concentration |
| chemical compound, drug | Hypoxanthine-Aminopterin-Thymidine; HAT | Thermo Fisher Scientific | 21060017 | Used at 1X final concentration |
| chemical compound, drug | Geneticin; G418 | Thermo Fisher Scientific | 10131027 | Used at 400 ug/mL final concentration |
| other | Ad-CMV-Cre | Vector Biolabs | 1045 | Adenovirus expressing Cre recombinase for excision of SEPT cassette from ExoI-targeted cell lines |
| software, algorithm | BWA-MEM v0.7.8 | PMID: 19451168 | NA | Used to align reads to human reference |
| software, algorithm | Picard v1.108 | Broad Institute; https://broadinstitute.github.io/picard/. | NA | Identify duplicate reads |
| software, algorithm | The Genome Analysis Toolkit (GATK) v2.8.1 | PMCID: PMC2928508 | NA | locally realign reads to known indels and recalibrate base quality scores |
| software, algorithm | MuTect v1.1.4 | PMCID: PMC3833702 | NA | Identiy somatic point mutations between the tumour and matched normal |
| other | WES/WGS raw sequencing data | This paper | NCBI GEO Accession: PRJNA327240 | Raw FASTQ files for WES and WGS performed in this study |

## Materials

Trypsin-EDTA was from Life Technologies and Geneticin was from Invitrogen. Antibodies against Mlh1 (mouse α-human Mlh1, G168-728) and β-actin (mouse α-human beta-actin, A1978) were from Pharmingen and Sigma, respectively.

## Cell culture

The human colorectal cancer cell line HCT-116 (a kind gift from Dr. Prescott Deininger) was grown in HyClone MEM/EBSS (Thermo Scientific) supplemented with 10% fetal bovine serum (Atlanta Biologicals), 1% sodium pyruvate (Invitrogen) and 1% MEM-NEAA (Invitrogen). The HCT-116 cells used in this study were validated via analysis of genome-wide mutation signature, microsatellite instability and biomarker. HCT-116 cells lack Mlh1 resulting in a well-characterized MSI phenotype (*Lynch et al., 1993*; *Parsons et al., 1993*; *Boland and Goel, 2010*). They further have a unique mutational spectrum that can be evaluated via next-generation sequencing (*Abaan et al., 2013*). Western blot analyses (*Figure 3A*) showed a lack of Mlh1 protein. The mutation spectrum from our whole-exome sequencing of HCT-116 cells (*Figure 2A* and *Figure 2—figure supplement 1*) is identical with that reported by Abaan

(*Abaan et al., 2013*). Lastly, we performed microsatellite stability analysis in our HCT116 cells at five mononucleotide homopolymeric run loci (NR27, NR21, NR24, BAT25, BAT26) using capillary electrophoresis, which showed instability at these loci providing a phenotypic readout consistent with the lack of Mlh1 expression in our cells (data not shown). The HCT-116 cell line is also not in the 488 commonly misidentified cell lines from the most recent ICLAS database (Version 8.0) and tested negative for mycoplasma.

## Generation of targeting constructs

In order to target the proofreading inactivating mutations to the POLE locus in vivo, we used rAAV with a synthetic exon promoter trap (*Rago et al., 2007*). A 1045 bp fragment containing POLE exons 7 and 8 along with intron 7 (termed HA1) was PCR amplified from HCT-116 genomic DNA using primers designed to add unique NotI and SacI sites to the 5′ and 3′ ends, respectively. A 1057 bp fragment containing exons 9, 10 and 11 along with introns 9 and 10 (termed HA2) was PCR amplified from HCT-116 genomic DNA using primers designed to add unique EcoRI and NotI sites to the 5′ and 3′ ends, respectively. Both HA1 and HA2 were first cloned into pCR-TOPO and sequence verified. The catalytic exonuclease DIE residues located in HA2 (exon 9) were changed to AIA using site-directed mutagenesis and sequence verified. The Pol ε rAAV shuttle vector was assembled by four-way ligation using the restriction enzyme-digested gene-specific HA1 and HA2 fragments, along with the SEPT/loxP cassette digested with NotI-EcoRI and the ITR-containing pAAV shuttle vector digested with NotI (SEPT/loxP cassette and pAAV shuttle vectors were kind gifts of Dr. Fred Bunz, Johns Hopkins University). The Exo-targeting vector was used to package high-titer ($1.6 \times 10^6$ PFU/ml) recombinant adeno-associated virus into AAV2 serotype capsids.

## Gene targeting and isolation of recombinant cell lines

Cells were grown in 100 mm dishes and infected with rAAV when ~75–80% confluent. At the time of infection, cells were washed with 1x Hanks buffered saline solution (Invitrogen) before adding 3 ml of media containing 75 µl of a 1:250 dilution of rAAV lysate. 3 hr after infection an additional 6 ml of media was added to plates and allowed to incubate at 37°C for 48 hr. After 48 hr, media was changed and Geneticin was added to a final concentration of 400 µg/ml. Plates were then incubated under selection for an additional 14 days. At the end of the selection period, colonies from plates were isolated using glass cloning rings and 0.05% trypsin (Invitrogen) was used to transfer colonies to 6-well plates for subsequent expansion. Genomic DNA was extracted from expanded clones using DNeasy Blood and Tissue kit (QIAgen) according to the manufacturer's protocol and eluted in 100 µl of elution buffer. Locus-specific integration was assessed by PCR using a primer that annealed outside the homology region and another that annealed within the *neo* cassette.

## Cre-mediated excision

To remove the SEPT cassette from correctly targeted clones, cells were infected in a 25 cm$^2$ flask with adenovirus that expresses the Cre recombinase ($1.0 \times 10^6$ PFU/ml, Vector Biolabs, Philadelphia, PA). Cells were plated at a limiting dilution in nonselective medium 24 hr after infection. 12 days after infection, single cell colonies were plated in duplicate and geneticin was added to one set of wells at a final concentration of 400 µg/ml to test for sensitivity. During this time, genomic DNA was extracted as previously described and screened using primers that annealed across both homology arms. PCR products were digested with SacI to distinguish between the wild type and recombinant locus.

## Southern blot analysis

Genomic DNA was harvested from the knock-in cell lines using the DNeasy Blood and Tissue Kit (Qiagen), and double digested with SacI and SalI. Hoechst fluorimetry was used to determine the concentration of DNA samples for accurate loading of samples. 4 µg of each sample was run on a 0.8% agarose gel in TBE. DNA was transferred to Hybond N + membrane (Amersham), blotted with a probe to HA2 at 65°C overnight, and washed at 65°C. To make the probe, a 300 bp sequence was amplified from the HA2-pCR-TOPO clone using the primers: 5′-GCATCTGCCCCACTGTTAGT-3′ and 5′-CTCCCTGTTGGTGATGAGGT-3′. The PCR product was labeled using the Prime-It II Random Primer Labeling Kit (Agilent) and α-$^{32}$P-dCTP (Perkin Elmer). Membrane was blocked in Denhardt's

pre-hybridization buffer [6x SSC, 0.5% SDS, 0.1% Ficoll 70, 0.1% Ficoll 400, 0.2% PVP, and 0.2%] at 65°C for 1 hr. The probe was added to hybridization buffer [6x SSC, 0.5% SDS, and 10% Dextran Sulfate] and incubated overnight at 65°C. To wash off excess probe, the blot was washed for 2 × 15 min washes in wash 1 [10x SSC, 0.5% SDS], 2 × 15 min washes in wash 2 [1x SSC, 1% SDS], and 2 × 30 min washes in wash 3 [0.1x SSC, 1% SDS]. The gel was exposed to a PhosphorImage screen and scanned on a Typhoon Imager.

## Purification of human Pol ε

An expression vector encoding residues 1–1189 of the catalytic subunit of human Pol ε containing the D275A/E277A substitution was prepared as described (*Korona et al., 2011*). Briefly, the human Pol ε was coexpressed in autoinduction medium with pRK603, which allows coexpression of TEV protease, at 25°C until the culture was saturated. Peak fractions from the HisTrap column were pooled, dialyzed into 50 mM HEPES, pH 7.5, 1 mM DTT, 5% glycerol and bound to SP sepharose. Bound protein was eluted with a 0–1 M with NaCl gradient. Peak fractions were pooled, dialyzed into 50 mM Tris, pH 7.5, 1 mM DTT, 5% glycerol, 100 mM NaCl and bound to Q Sepharose. Bound protein was eluted with a 100 mM–M M NaCl gradient. Peak fractions were pooled, concentrated and passed through a pre-equilibrated Superdex200 size exclusion column. Fractions containing the purified 140 kDa protein were pooled, dialyzed into 50 mM Tris, pH 8.0, 1 mM DTT, 5% glycerol and aliquots were frozen and stored at −80°C.

## TCT→TAT in vitro error rate

We previously reported that the lacZ forward mutation assay template lacks sites at which TCT→TAT transversions are phenotypically detectable (*Shlien et al., 2015*). To overcome this limitation we previously made a reversion substrate that reports only this mutation by using site-directed mutagenesis to change $A_{-11}$ to $C_{-11}$. Double-stranded M13mp2 DNA containing the $TC_{-11}T$ sequence was used as a substrate in reactions containing 0.15 nM DNA, 50 mM Tris-Cl, pH 7.4, 8 mM MgCl2, 2 mM DTT, 100 µg/ml BSA, 10% glycerol, 250 µM dNTPs and 1.5 nM Pol ε at 37°C. Completely filled product was transfected into Escherichia coli cells, which were used to determine the frequency of dark blue revertant plaques that occurred as a result of TCT→TAT transversions arising during DNA synthesis. In this assay, accurate DNA synthesis yields colorless plaques. Error rates were calculated according to the following equation: error rate (per nucleotide synthesized) = ((number of mutants of a particular class) × (mutant frequency)) / ((number of mutations sequenced) × (0.6) × (number of detectable sites)).

## Mlh1 lentivirus construction

Mlh1 ORF was PCR amplified using the pCMV-XL5-Mlh1 vector (kindly provided by Victoria Belancio, Tulane University), forward and reverse primers (fwd 5′-TCGACTCGAGTCCACCATGTCGTTCG TGGCAGG-3′; rev 5′-TCGAGGATCCGTTACTTAACACCTCTCAAAGAC-3′) and Q5 DNA polymerase (NEB). After gel purification, dA was added to the 3′ ends with Taq and the Mlh1 ORF was cloned into pLenti6.3/V5-TOPO (Invitrogen). Mlh1 was found to have a common I219V SNP that does not affect Mlh1 function (*Plotz et al., 2008*). Mlh1 Lentiviral particles were made using the ViraPower Lentiral Expression System (Invitrogen). Briefly, 293FT cells were transfected with pLenti6.3/V5-TOPO-Mlh1 and a mixture of plasmids encoding lentiviral packaging factors. Viral supernatant was harvested 48 hr after transfection, filter sterilized and stored in aliquots at −80°C. After titering, HCT-116 cells were transduced with Mlh1 lentivirus at MOI of 1.0. Cells were selected for 1 week in 10 µg/ml blasticidin. Blasticidin-resistant clones were identified and cells were harvested, lysed and probed by Western blot (mouse α-human Mlh1, G168-728, Pharmingen) to confirm Mlh1 expression.

## Mutation rate and mutant frequency measurements

Prior to mutation rate measurements, preexisting HPRT1 mutants were eliminated from cell populations by incubating cells in HAT medium (1x Hypoxanthine-Aminopterin-Thymidine) for five passages. For each cell line analyzed, 500 cells were seeded and grown to confluence in 12 wells across two 6-well plates. Cells from one well were harvested and counted to estimate cell number in the remaining 11 wells. For mutation rate measurement, 500 cells from each of the remaining eleven wells were seeded per dish in 3 × 100 mm dishes in media lacking 6-TG to be used to measure

plating efficiency. At the same time, $5 \times 10^5$ cells from each of the remaining eleven wells were plated in $5 \times 100$ mm dishes in media containing 6-TG. After 7 days, colonies on the plating efficiency wells were stained with crystal violet and counted. After 12–14 days, the 6-TG resistant colonies were also stained with crystal violet and counted. Mutation rate was calculated using the Ma-Sandri-Sarkar Maximum Likelihood Estimator (MSS-MLE) method (*Rosche and Foster, 2000*).

For mutant frequency measurement, 500 cells per clone were seeded in duplicate in 6-well plates in media lacking 6-TG and allowed to grow for 5–7 days to determine plating efficiency. The remaining wells were seeded with $5 \times 10^4$ cells in media containing 6-TG and allowed to grow for 12–14 days. After the indicated time, colonies were stained with crystal violet and counted. Mutant frequency was calculated by the following equation: (# 6-TG resistant colonies) / ([(# colonies scored$_{PE}$)/ (# cells seeded$_{PE}$)] x (# cells seeded$_{6-TG}$)). PE refers to plating efficiency. Colonies were defined as $\geq$50 cells.

HCT116 and HCT116 + Mlh1 cells were seeded into T75 flasks and grown at 37°C/5% $CO_2$ until 80% confluency was reached. Cells were counted using the Countess Automated Cell Counter (Invitrogen) and $1 \times 10^6$ cells were seeded into new T75 flasks and incubated until 80% confluency was reached. The above protocol was repeated at regular intervals (3–4 days) and population doubling (PDL) numbers calculated using the following equation: PDL = [ln($N_t$)-ln($N_0$*PE)]/ln2. $N_t$ = Number of viable cells counted after passage; $N_0$ = Number of cells seeded prior to passage; PE = plating efficiency. At PDL $\sim$ 6, 44 and 69 cells were trypsinized and counted. For mutant frequency measurement, 300 cells were seeded into each of $3 \times 100$ mm dishes in media lacking 6-TG to be used to measure plating efficiency. Concurrently, $2 \times 10^5$ cells were seeded into each of $10 \times 100$ mm dishes in media supplemented with 6-TG to a final concentration of 5 μg/mL. After 7 days, colonies on the plating efficiency dishes were stained with crystal violet and counted. After 12–14 days, 6-TG resistant colonies were isolated using glass cloning rings and 0.05% trypsin and transferred into 24-well plates for expansion and RNA isolation. Additionally, at the above PDLs an aliquot of cells were harvested, lysed and probed by Western blot (mouse α-human Mlh1, G168-15, Abcam) to confirm maintenance of Mlh1 expression.

Genomic per base pair mutation rates ($\mu_{BS}$) were calculated using the method of Drake (*Drake, 1991*) with modifications as applied in Lynch (*Lynch, 2010*). The equation used was: $\mu_{BS} = (\mu_L \bullet f_T \bullet f_{BS}) / (L \bullet f_L \bullet [x (n_m + n_n)/n_n])$, where $\mu_L$ is the measured mutation rate at the HPRT1 reporter gene, $f_T$ is the fraction of mutants found after sequencing, $f_{BS}$ is the fraction of mutations due to base pair substitutions, L is the length (in nt) of the reporter gene, $f_L$ is the fraction of HPRT1 that gives rise to detectable mutations, x is the fraction of mutations that would give rise to chain terminator mutations, $n_m$ is the observed number of missense mutations and $n_n$ is the observed number of nonsense mutations. We used 126 HPRT1 mutations from three independent studies (*Bhattacharyya et al., 1995*; *Glaab and Tindall, 1997*; *Ohzeki et al., 1997*) to calculate $\mu_{BS}$. The values used were: $f_T$ = 1.0, $f_{BS}$ = 79/126 = 0.627; L = 627 nt; $f_L$ = 1; x = 3/64 = 0.047; $n_m$ = 74; $n_n$ = 5. The $\mu_L$ value for Pol ε mutant cell lines was determined empirically using fluctuation analysis.

## HPRT1 sequencing

Total RNA was isolated using the Qiagen RNeasy kit (Qiagen) according to the manufacturer's protocol. RT-PCR was performed with SuperScript III Reverse Transcriptase (Invitrogen) according to the manufacturer's protocol using 1 μg of RNA as a template. Primer-specific cDNA was amplified for 32 cycles at an annealing temperature of 60°C using the following HPRT1 primers: 123(fwd) CTTCC TCCTCCTGAGCAGTC and 1041 (rev) GCCCAAAGGGAACTGATAGTC. From the HPRT1 sequencing of 6-TG resistant colonies, one clone was found to have exon 2 completely deleted. Exon deletions in HPRT1 have been shown to be caused by splice site mutations (*Bhattacharyya et al., 1995*). We therefore amplified exon 2 and its flanking region from genomic DNA prepared from the appropriate clone using the following primers: Forward: TTGTTTTCTTACATAATTCATTATCATACC; Reverse: TTACTTTGTTCTGGTCCCTACAGAG.

## Whole genome and exome sequencing

Next generation sequencing was performed as per the published protocols. Whole genome sequencing (WGS) was performed on an Illumina HiSeq Xten instrument with libraries prepared using the manufacturer's TruSeq Nano DNA Library Prep kit and sequenced to a depth of 36.1x. For

exome sequencing, DNA was enriched using Agilent SureSelect Human Exome Library Preparation V5 kit, then sequenced to a depth of 101.38x (96.61x-108.19x).

## Substitution detection from next generation sequenced data

All samples were processed from raw reads (FASTQ files) from paired end libraries. The reads were aligned to the human reference (GRCh37 with decoy sequences) using BWA-MEM v0.7.8 (*Li and Durbin, 2009*). Duplicate reads were identified and marked using Picard v1.108 (https://broadinstitute.github.io/picard/). The Genome Analysis Toolkit (GATK) v2.8.1 (*McKenna et al., 2010*) was used to locally realign reads to known indels and recalibrate base quality scores. Quality metrics were generated from the final BAM files to ensure high quality alignment. This includes:

- average coverage >90 x in whole exome data (*Figure 2—figure supplement 6* Mean_Coverage_Per_Sample.pdf)
- alignment rate to the reference genome >99% across whole exome data (*Figure 2—figure supplement 7* Proportion_of_properly_paired_reads.pdf) with >60M reads per sample (*Figure 2—figure supplement 8* Total_Reads_Exome.pdf)
- >90% of bases in the genome at >20 x coverage and >90% of bases in the exome at >30 x coverage (*Figure 2—figure supplement 9*)

Limitations in the genome due to low-complexity regions and incomplete areas in the genome (*Li, 2014*) prevent proper alignment resulting in sources of error.

Somatic point mutations between the tumour and matched normal were identified using MuTect v1.1.4 (*Cibulskis et al., 2013*). In addition, we used MuTect v1.1.4 in single sample mode to detect all mutations in each sample. All mutations were annotated using ANNOVAR v20130823 (*Wang et al., 2010*). Subsequent filtering was performed to reduce potential false positives and allow only high confidence mutations in the dataset using a custom R package (ShlienLab.Core.SNV v0.09). Mutations were retained if they met the following criteria:

- not identified in common mutation databases including: dbSNP (138), 1000 genomes (1000g2012feb), complete genomics (CG69), Exome sequencing project (ESP 6500si)
- for exome data, must have at least 20x normal and 30x tumour
- for WGS data, must have at least 10x normal and 10x tumour (*Figure 2—figure supplement 10*)

To investigate the quality of somatic mutations, we also identified key metrics including:

- Average alternate base quality to reference base quality of ~1.0 (*Figure 2—figure supplement 11*, mean_ratio_tumour_alt_ref_base_quality.pdf)

## Data access

DNA sequencing data from this study have been submitted to the NCBI Gene Expression Omnibus (GEO; http://www.ncbi.nlm.nih.gov/geo/) under accession number PRJNA327240.

# Acknowledgements

The authors would like to thank Dr. Fred Bunz (John Hopkins University) and Drs. Prescott Deininger and Victoria Belancio (Tulane University) for the kind sharing of reagents. Thanks are also due to Christine McBride for her contribution to the rAAV construction. Additionally, the authors would like to thank Dr. Art Lustig and Dr. Stuart Linn for insightful comments and advice.

# Additional information

## Funding

| Funder | Grant reference number | Author |
|---|---|---|
| Tulane University | Stem Cell and Regenerative Medicine Faculty Grant | Bruce A Bunnell |
| National Institute of Environmental Health Sciences | NIH R01ES028271 | Zachary F Pursell |

| National Institute of Environmental Health Sciences | NIH R56ES026821 | Zachary F Pursell |
|---|---|---|
| National Institute of Environmental Health Sciences | NIH R00 ES016780 | Zachary F Pursell |
| National Institute of Environmental Health Sciences | NIH P20 RR020152 | Zachary F Pursell |

The funders had no role in study design, data collection and interpretation, or the decision to submit the work for publication.

### Author contributions

Karl P Hodel, Conceptualization, Formal analysis, Validation, Investigation, Visualization, Methodology, Writing—original draft, Writing—review and editing; Richard de Borja, Data curation, Software, Formal analysis, Investigation, Visualization, Methodology, Writing—review and editing; Erin E Henninger, Conceptualization, Formal analysis, Investigation, Visualization, Methodology, Writing—review and editing; Brittany B Campbell, Formal analysis, Validation, Investigation, Visualization, Methodology, Writing—review and editing; Nathan Ungerleider, Resources, Software, Formal analysis, Investigation, Visualization, Methodology; Nicholas Light, Data curation, Software, Formal analysis, Investigation, Methodology, Writing—review and editing; Tong Wu, Kimberly G LeCompte, Bruce A Bunnell, Investigation, Methodology, Writing—review and editing; A Yasemin Goksenin, Investigation, Methodology; Uri Tabori, Conceptualization, Formal analysis, Supervision, Funding acquisition, Validation, Methodology, Writing—review and editing; Adam Shlien, Conceptualization, Resources, Data curation, Software, Formal analysis, Supervision, Funding acquisition, Validation, Investigation, Visualization, Methodology, Writing—review and editing; Zachary F Pursell, Conceptualization, Resources, Formal analysis, Supervision, Funding acquisition, Validation, Investigation, Visualization, Methodology, Writing—original draft, Project administration, Writing—review and editing

### Author ORCIDs

Zachary F Pursell http://orcid.org/0000-0001-5871-7192

### Decision letter and Author response

Decision letter https://doi.org/10.7554/eLife.32692.033
Author response https://doi.org/10.7554/eLife.32692.034

# Additional files

### Supplementary files

• Transparent reporting form
DOI: https://doi.org/10.7554/eLife.32692.027

### Major datasets

The following dataset was generated:

| Author(s) | Year | Dataset title | Dataset URL | Database, license, and accessibility information |
|---|---|---|---|---|
| Hodel KP | 2016 | Homo sapiens Raw sequence reads - BioProject | https://www.ncbi.nlm.nih.gov/bioproject/?term=PRJNA327240 | Publicly available at NCBI BioProject (Accession no. PRJNA327240) |

The following previously published dataset was used:

| Author(s) | Year | Dataset title | Dataset URL | Database, license, and accessibility information |
|---|---|---|---|---|
| Abaan OD | 2013 | NCI-60 mutation dataset | https://cancer.sanger.ac.uk/cosmic/download | Publicly available in the Catalogue Of Somatic Mutations In Cancer (COSMIC) database at the Wellcome Sanger Institute (file labelled: 'CosmicMutantExport.tsv.gz') |

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
