## [Decision Letter]

Thank you for submitting your article "Explosive mutation accumulation triggered by human Pol ε proofreading-deficiency is driven by a complex mechanism" for consideration by *eLife*. Your article has been reviewed by three peer reviewers, one of whom is a member of our Board of Reviewing Editors and the evaluation has been overseen by Sean Morrison as the Senior Editor. The reviewers have opted to remain anonymous.

The reviewers have discussed the reviews with one another and the Reviewing Editor has drafted this decision to help you prepare a revised submission.

Summary:

This manuscript discusses the interrelationship between Pol epsilon proofreading deficiencies and defects in mismatch repair and their effects on mutational frequencies. Hodel et al. use human cell lines with different combinations of defects in proofreading and MMR and perform sequencing to compare their mutational spectra with that of tumors from patients with biallelic mismatch repair deficiency. The authors have chosen to study the D275A/E277A double substitution in human Pol epsilon, which has not been found in cancers, but is a well-characterized mutant in both yeast and mouse. Thus, it is a good choice for this type of study because it is catalytically dead and the results can be compared to observations in model organisms. It is already known that tumors with exonuclease deficient Pol epsilon and mismatch repair have a very high mutation load. However, the mutation rates cannot be measured in tumors, only the frequency. Here the authors determine the mutation rates in both wt/exo-deficient POLE cells with or without functional mismatch repair. The authors find that unaffected mismatch repair is able to entirely suppress mutation accumulation caused by deficiencies on proofreading, suggesting that additional defects are required to explain mutation acquisition in tumors containing proofreading-deficient Pol epsilon.

Essential revisions:

1) This is a comprehensive study that provides insight into the mechanisms underlying mutation accumulation in a human cell line, with a direct relevance to clinically relevant cancers. The key issue that was raised during review of the manuscript is that the main outcomes of the work are poorly articulated in the context of the existing literature and the field's current understanding of the impact of proofreading deficiencies on mutation accumulation. In particular, the authors should better rationalise their study design using mono-allelic Pol epsilon exo minus and contrast it with previous work on model systems that were bi-allelic in proofreading deficiency. The way the manuscript currently reads, it is not clear why only one allele of Pol epsilon is inactivated. Relatedly, they should better articulate the relevance of their work to the clinical situation compared to previous work on model systems.

2) Throughout the manuscript, several graphs and values do not contain error bars/margins. This is a significant concern, with the interpretation of the results critically relying on an understanding of the robustness of the experimental values.

3) The authors suggest that Pol epsilon exo-minus tumors lacking all mismatch repair function are microsatellite stable. A complete lack of mismatch repair typically leads to microsatellite instability and as a result the microsatellite stable phenotype was seen as surprising by one of the reviewers. The authors should provide a rationalisation for this observation.

4) The Title of the manuscript is misleading. The authors claim the mutation accumulation driven by a pol epsilon deficiency is "complex", yet the data presented is convincing that its driven by a mismatch repair deficiency.

---

## [Author Response]

Essential revisions:1) This is a comprehensive study that provides insight into the mechanisms underlying mutation accumulation in a human cell line, with a direct relevance to clinically relevant cancers. The key issue that was raised during review of the manuscript is that the main outcomes of the work are poorly articulated in the context of the existing literature and the field's current understanding of the impact of proofreading deficiencies on mutation accumulation. In particular, the authors should better rationalise their study design using mono-allelic Pol epsilon exo minus and contrast it with previous work on model systems that were bi-allelic in proofreading deficiency. The way the manuscript currently reads, it is not clear why only one allele of Pol epsilon is inactivated. Relatedly, they should better articulate the relevance of their work to the clinical situation compared to previous work on model systems.

We thank the reviewers for their critical assessment of this manuscript. We chose to inactivate a single allele of Pol epsilon so as to be consistent with what is seen in Pol epsilon patient tumors as well as inherited germline mutations in Pol epsilon. In almost all reported cases of hypermutated POLE tumors, the Pol epsilon mutation is heterozygous with no reported loss of heterozygosity (LOH). Cell lines created from these tumors are also heterozygous for the Pol epsilon mutation. Since the wild type allele is presumed to be expressed in these tumors, important questions regarding full or partial dominance of the mutant allele remain unaddressed. Indeed, results from yeast suggest that the mutator phenotype arises in the diploid heterozygous mutant. Thus, constructing a mono-allelic system is critical to understanding the mutagenic and clinical situation in these patients. Our previous biochemical work demonstrated that the Pol epsilon mutant studied here is an appropriate model to study in cells. Specifically, the proofreading exonuclease activity defects of the current mutant are comparable to those of several patient-derived mutations studied. We regret that we did not make this more clear and have strengthened this rationale in the introduction.

POLE tumors are clinically characterized by several criteria, including being hypermutant, being heterozygous with no LOH, having a unique mutational signature and being microsatellite stable (MSS). (There is some discrepancy regarding this last criteria that we discuss in greater detail later in this response.) The current study in part aims to help better understand the mechanisms responsible for these clinical characteristics in the context of the rich history of work on Polymerase epsilon and δ mutants in model systems. Our results establish that loss of 3’-5’ exonuclease activity in one allele is sufficient to drive several of these clinical criteria (the hypermutant phenotype and most of the unique mutational signature) when present in the heterozygous state, provided MMR is non-functional. These results are consistent with what is seen for mutation rates in Pole^wt/exo-^ MMR-deficient yeast and in the special case of bMMRD patients who develop spontaneous POLE heterozygous mutations in the complete absence of MMR. Perhaps most importantly, we believe that the lack of a hypermutant phenotype when MMR is fully restored in our Pole^wt/exo-^ monoallelic cell model helps explain the lack of tumors observed in the equivalent Pole^wt/exo-^ monoallelic mouse model system. We believe that functional MMR in these mice, Pole^wt/exo-^ and MMR-proficient, restrains the development of the unique hypermutant phenotype, thus preventing tumor development. These observations support the idea that the pathogenicity of POLE mutants may be more directly related to some additional defect in the mutant polymerase (Barbari and Shcherbakova, 2017). The best example of this is the much stronger effects on mutation rate in yeast for the P286R equivalent mutation than for the D275A/E275A mutant equivalent (Kane and Shcherbakova, 2014). This idea is further supported by studies in yeast on Pol δ mutants that showed differing effects on mutagenesis for engineered exo-domain mutants vs mutants selected for mutagenesis over generations (Murphy et al., 2006, Genome). Based on our work and those of model systems, we believe that a subset of the clinical criteria arise solely from loss of a single allele of polymerase proofreading, while the full set of clinical characteristics are dependent on the particular POLE allele. The mutations found in POLE tumors by definition satisfy all criteria. This idea can be tested directly in our cell model system by comparing patient-derived POLE mutations directly to the mutant used in the current studies. We have made every effort to better articulate these views and observations throughout the text.

2) Throughout the manuscript, several graphs and values do not contain error bars/margins. This is a significant concern, with the interpretation of the results critically relying on an understanding of the robustness of the experimental values.

We thank the reviewers for this critical observation. We have now provided the relevant information where applicable:

Figure 1-values have been added to the figure legend.

Figure 1: Error bars have been added to the figure. We also now report the mean, SEM, n and exact p-value in the figure legend. BPS and FS error rate uncertainties were propagated using the following formula (ER, Error Rate): 𝜎_ER_ = ER * sqrt((𝜎_Mutation Class Proportion_/Mutation Class Proportion)^2^+(𝜎_Mutation Rate_/Mutation Rate)^2^).

Figure 1: We have removed Figure 1 from the manuscript as there are not enough data to draw statistically reliable conclusions concerning the TCT>TAT Error Rate at the HPRT1 locus.

Figure 1 (which is now Figure 1 in the revised manuscript): We now compare enzymatic error rates for in vitro TCT>TAT reversion using a Chi-squared analysis. This is consistent with the statistical methods we reported previously (Shlien et al., 2014). We also provide comparison to clinically relevant mutant Pol ε. p-values and the chi-square statistic are provided in the table.

Figure 2: The data presented are proportions of counts observed for each BPS class in a single population and are thus treated as a single experiment and analyzed via chi-square analysis. We have amended the figure legend to contain the chi-square statistic and p-values.

Figure 2: We have now indicated significance for the indicated Pol ε hot spot mutations and state the chi-square statistic and p-value in the figure legend.

Figure 3: We now report the uncertainty values shown in the figure for the Mlh1-corrected clones in the legend in addition to the mean and p-values.

Figure 4: Summary statistics are now made available in the figure legend.

Figure 4: The p value was previously reported as p=0.0006. The actual value, p=0.0002, is now correctly reported.

3) The authors suggest that Pol epsilon exo-minus tumors lacking all mismatch repair function are microsatellite stable. A complete lack of mismatch repair typically leads to microsatellite instability and as a result the microsatellite stable phenotype was seen as surprising by one of the reviewers. The authors should provide a rationalisation for this observation.

We did not mean to suggest that Pol epsilon exo-deficient tumors lacking all MMR function are microsatellite stable. It is clear from the literature that tumors initiating from a complete inactivation of MMR give rise to readily detectable MSI phenotype, which is generally MSI-H. Even in bMMRD patients with fully biallelic exonuclease-proficient Pol ε clearly show MSI in tumors. We apologize for not making this more clear. We have rewritten the discussion to more clearly reflect this and the following discussion. Rather, we are intrigued by the subset of POLE hypermutant tumors that were initially clinically diagnosed as MSI-L and MSI-H by PCR analyses and had mutations in various mismatch repair genes. Analysis of the WES/WGS data showed no evidence of instability in 46 homonucleotide runs (Shinbrot, 2013). We envision three testable possibilities to explain the discrepancy in the clinical data: (1) POLE-induced mutations arise independently of MMR; (2) MMR is saturated by excessive mutant POLE-induced replication errors; (3) MMR is transiently suppressed in POLE tumors for some unknown period of time.

In the first case, MMR would be fully functional throughout tumor progression. The discrepancy in MSS vs MSI-L/-H POLE clinical data could be explained by false positives in the initial data. Indeed, microsatellite marker sensitivity is sensitive to the specific sequence being examined. In fact, one of the most variable MSI markers in common use, the dinucleotide D5S346 marker [Hile and Eckert, 2013], was used in both of the large cohort next generation TCGA studies on colorectal and endometrial cancers [TCGA, 2012 and Kandoth, 2013]. This case is further supported by the re-analysis demonstrating stability in a panel of over 40 mononucleotide sequences by NGS [Shinbrot, 2013]. As no mutagenesis was observed using our exo-deficient mutant, an allele-specific mutagenesis would explain this case. Careful re-examination of more complex microsatellite sequences in the tumor data would help resolve this.

The second case would be consistent with studies in extreme mutators in model systems. However, the exonuclease-deficient mutant studied here clearly does not saturate functional mismatch repair in the cell model system. This further underscores the possibility that allele-specific differences in POLE mutants could be driving the observed mutagenesis. As discussed above, studies with the P286R mutant in yeast support this model and need to be corroborated in human cell models. These studies are ongoing in our laboratory.

The third case would suggest that MMR is functionally compromised for some period of time. This would be consistent with the original clinical diagnosis of MSI-L/-H for a subset of POLE tumors and could resemble a situation more resembling alternative-MSI. A-MSI occurs in certain cases of MSI-L caused by instability at more complex repeats; and in cases of EMAST, or Elevated Microsatellite Alterations at Selected Tetranucleotide repeats. These cases are generally not seen in tumors with complete MMR inactivation (e.g. MLH1 or MSH2 loss). They occur more frequently from loss of PMS2, MSH3, Pol kappa or even increased environmental DNA damage. This case would also be consistent with a number of reports in the literature showing that certain types of MMR suppression are reversible. These are generally via chemical treatments, including adriamycin, Cadmium, Zinc and a DNA base analog (Larson et al., 2000; Banerjee et al., 2005; Negishi et al., 2002). Appearance of a true MSI phenotype would be dependent on the nature and length of exposure, as well as the microsatellite sequence. One interesting set of experiments demonstrates the highly variable nature of observable MSI phenotype when MMR is transiently suppressed (Koole et al., 2013). Using a fluorescent reversion reporter and siRNA against MSH2, they showed that even the most sensitive reporter used (23 guanines) required two rounds of knockdown and two weeks of passaging before 4% of cells showed detectable MSI at the reporter. This would be below the detection level of all but the highest depth WES/WGS. Interestingly, in this system no MSI was detectable for adenine or thymine runs, or for runs less than 11nt. This is a very interesting area of future studies as patient exposures are highly variable over a lifetime.

4) The title of the manuscript is misleading. The authors claim the mutation accumulation driven by a pol epsilon deficiency is "complex", yet the data presented is convincing that its driven by a mismatch repair deficiency.

While the data are convincing that mismatch repair deficiency is driving mutation accumulation with the exonuclease-deficient allele in the current study, our title was chosen to reflect the possibility raised by studies in yeast that the cancer mutants are somehow intrinsically different with respect to mutagenesis in vivo (Kane and Shcherbakova, 2014). This remains speculative in the absence of any functional data using human cells. We have revised our title to more accurately reflect the data as presented.